# Fair Hierarchical Clustering

**Sara Ahmadian**
Google
sahmadian@google.com

**Alessandro Epasto**
Google
aepasto@google.com

**Marina Knittel**
University of Maryland
mknittel@cs.umd.edu

**Ravi Kumar**
Google
ravi.k53@gmail.com

**Mohammad Mahdian**
Google
mahdian@google.com

**Benjamin Moseley**
Carnegie Mellon University
moseleyb@andrew.cmu.edu

**Philip Pham**
Google
phillypham@google.com

**Sergei Vassilvitskii**
Google
sergeiv@google.com

**Yuyan Wang**
Carnegie Mellon University
yuyanw@andrew.cmu.edu

## Abstract

As machine learning has become more prevalent, researchers have begun to recognize the necessity of ensuring machine learning systems are fair. Recently, there has been an interest in defining a notion of fairness that mitigates over-representation in traditional clustering.

In this paper we extend this notion to hierarchical clustering, where the goal is to recursively partition the data to optimize a specific objective. For various natural objectives, we obtain simple, efficient algorithms to find a provably good fair hierarchical clustering. Empirically, we show that our algorithms can find a fair hierarchical clustering, with only a negligible loss in the objective.

## 1 Introduction

Algorithms and machine learned models are increasingly used to assist in decision making on a wide range of issues, from mortgage approval to court sentencing recommendations [28]. It is clearly undesirable, and in many cases illegal, for models to be biased to groups, for instance to discriminate on the basis of race or religion. Ensuring that there is no bias is not as easy as removing these protected categories from the data. Even without them being explicitly listed, the correlation between sensitive features and the rest of the training data may still cause the algorithm to be biased. This has led to an emergent literature on computing provably fair outcomes (see the book [7]).

The prominence of clustering in data analysis, combined with its use for data segmentation, feature engineering, and visualization makes it critical that efficient fair clustering methods are developed. There has been a flurry of recent results in the ML research community, proposing algorithms for fair *flat* clustering, i.e., partitioning a dataset into a set of disjoint clusters, as captured by K-CENTER, K-MEDIAN, K-MEANS, correlation clustering objectives [3, 4, 6, 8, 9, 14, 18, 24, 25, 30, 31]. However, the same issues affect *hierarchical clustering*, which is the problem we study.

The input to the hierarchical clustering problem is a set of data points, with pairwise similarity or dissimilarity scores. A hierarchical clustering is a tree, whose leaves correspond to the individual datapoints. Each internal node represents a cluster containing all the points in the leaves of its subtree. Naturally, the cluster at an internal node is the union of the clusters given by its children. Hierarchical clustering is widely used in data analysis [21], social networks [32, 34], and image/text organization [26].

Hierarchical clustering is frequently used for flat clustering when the number of clusters is a priori unknown. A hierarchical clustering yields a set of clusterings at different granularities that are consistent with each other. Therefore, in all clustering problems where fairness is desired but the number of clusters is unknown, fair hierarchical clustering is useful. As concrete examples, consider a set of news articles organized by a topic hierarchy, where we wish to ensure that no single source or view point is over-represented in a cluster; or a hierarchical division of a geographic area, where the sensitive attribute is gender or race, and we wish to ensure balance in every level of the hierarchy. There are many such problems that benefit from fair hierarchical clustering, motivating its study.

**Our contributions.** We initiate an algorithmic study of fair hierarchical clustering. We build on Dasgupta's seminal formal treatment of hierarchical clustering [20] and prove our results for the revenue [33], value [19], and cost [20] objectives in his framework.

To achieve fairness, we show how to extend the fairlets machinery, introduced by [16] and extended by [3], to this problem. We then investigate the complexity of finding a good fairlet decomposition, giving both strong computational lower bounds and polynomial time approximation algorithms.

Finally, we conclude with an empirical evaluation of our approach. We show that ignoring protected attributes when performing hierarchical clustering can lead to unfair clusters. On the other hand, adopting the fairlet framework in conjunction with the approximation algorithms we propose yields *fair* clusters with a *negligible* objective degradation.

**Related work.** Hierarchical clustering has received increased attention over the past few years. Dasgupta [20] developed a cost function objective for data sets with similarity scores, where similar points are encouraged to be clustered together lower in the tree. Cohen-Addad et al. [19] generalized these results into a class of optimization functions that possess other desirable properties and introduced their own value objective in the dissimilarity score context. In addition to validating their objective on inputs with known ground truth, they gave a theoretical justification for the average-linkage algorithm, one of the most popular algorithms used in practice, as a constant-factor approximation for value. Contemporaneously, Moseley and Wang [33] designed a revenue objective function based on the work of Dasgupta for point sets with similarity scores and showed the average-linkage algorithm is a constant approximation for this objective as well. This work was further improved by Charikar [13] who gave a tighter analysis of average-linkage for Euclidean data for this objective and [2, 5] who improved the approximation ratio in the general case.

In parallel to the new developments in algorithms for hierarchical clustering, there has been tremendous development in the area of fair machine learning. We refer the reader to a recent textbook [7] for a rich overview, and focus here on progress for fair clustering. Chierichetti et al. [16] first defined fairness for $k$-median and $k$-center clustering, and introduced the notion of *fairlets* to design efficient algorithms. Extensive research has focused on two topics: adapting the definition of fairness to broader contexts, and designing efficient algorithms for finding good fairlet decompositions. For the first topic, the fairness definition was extended to multiple values for the protected feature [3, 9, 35]. For the second topic, Backurs et al. [6] proposed a near-linear constant approximation algorithm for finding fairlets for $k$-median, Schmidt et al. [36] introduced a streaming algorithm for scalable computation of coresets for fair clustering, Kleindessner et al. [30] designed a linear time constant approximation algorithm for $k$-center where cluster centers are selected proportionally from a set of colors, Bercea et al. [9] developed methods for fair $k$-means, while Ahmadian et al. [4] and Ahmadi et al. [1] defined approximation algorithms for fair correlation clustering. Concurrently with our work, Chhabra et al. [15] introduced a possible approach to ensuring fairness in hierarchical clustering. However, their fairness definition differs from ours (in particular, they do not ensure that all levels of the tree are fair), and the methods they introduce are heuristic, without formal fairness or quality guarantees.

Beyond clustering, the same balance notion that we use has been utilized to capture fairness in other contexts, for instance: fair voting [10], fair optimization [17], as well as other problems [11].

## 2 Formulation

### 2.1 Objectives for hierarchical clustering

Let $G = (V, s)$ be an input instance, where $V$ is a set of $n$ data points, and $s : V^2 \to \mathbb{R}^{\geq 0}$ is a similarity function over vertex pairs. For two sets, $A, B \subseteq V$, we let $s(A, B) = \sum_{a \in A, b \in B} s(a, b)$

and $S(A) = \sum_{\{i,j\} \in A^2} s(i,j)$. For problems where the input is $G = (V, d)$, with $d$ a distance function, we define $d(A, B)$ and $d(A)$ similarly. We also consider the *vertex-weighted* versions of the problem, i.e. $G = (V, s, m)$ (or $G = (V, d, m)$), where $m : V \to \mathbb{Z}^+$ is a weight function on the vertices. The vertex-unweighted version can be interpreted as setting $m(i) = 1, \forall i \in V$. For $U \subseteq V$, we use the notation $m(U) = \sum_{i \in U} m(i)$.

A *hierarchical clustering* of $G$ is a tree whose leaves correspond to $V$ and whose internal vertices represent the merging of vertices (or clusters) into larger clusters until all data merges at the root. The goal of hierarchical clustering is to build a tree to optimize some objective.

To define these objectives formally, we need some notation. Let $T$ be a hierarchical clustering tree of $G$. For two leaves $i$ and $j$, we say $i \vee j$ is their least common ancestor. For an internal vertex $u$ in $T$, let $T[u]$ be the subtree in $T$ rooted at $u$. Let leaves($T[u]$) be the leaves of $T[u]$.

We consider three different objectives—*revenue*, *value*, and *cost*—based on the seminal framework of [20], and generalize them to the vertex-weighted case.

**Revenue.** Moseley and Wang [33] introduced the revenue objective for hierarchical clustering. Here the input instance is of the form $G = (V, s, m)$, where $s : V^2 \to \mathbb{R}^{\geq 0}$ is a *similarity* function.

**Definition 1** (Revenue). *The* revenue *(rev) of a tree $T$ for an instance $G = (V, s, m)$, where $s(\cdot, \cdot)$ denotes similarity between data points, is:* $\mathrm{rev}_G(T) = \sum_{i,j \in V} s(i,j) \cdot \big( m(V) - m(\mathrm{leaves}(T[i \vee j])) \big)$.

Note that in this definition, each weight is scaled by (the vertex-weight of) the non-leaves. The goal is to find a tree of maximum revenue. It is known that average-linkage is a $1/3$-approximation for vertex-unweighted revenue [33]; the state-of-the-art is a $0.585$-approximation [5].

As part of the analysis, there is an upper bound for the revenue objective [19, 33], which is easily extended to the vertex-weighted setting:

$$\mathrm{rev}_G(T) \leq \Big( m(V) - \min_{u,v \in V, u \neq v} m(\{u, v\}) \Big) \cdot s(V). \tag{1}$$

Note that in the vertex-unweighted case, the upper bound is just $(|V| - 2)s(V)$.

**Value.** A different objective was proposed by Cohen-Addad et al. [19], using distances instead of similarities. Let $G = (V, d, m)$, where $d : V^2 \to \mathbb{R}^{\geq 0}$ is a distance (or dissimilarity) function.

**Definition 2** (Value). *The* value *(val) of a tree $T$ for an instance $G = (V, d, m)$ where $d(\cdot, \cdot)$ denotes distance is:* $\mathrm{val}_G(T) = \sum_{i,j \in V} d(i,j) \cdot m(\mathrm{leaves}(T[i \vee j]))$.

As in revenue, we aim to find a hierarchical clustering to maximize value. Cohen-Addad et al. [19] showed that both average-linkage and a locally $\epsilon$-densest cut algorithm achieve a $2/3$-approximation for vertex-unweighted value. They also provided an upper bound for value, much like that in (1), which in the vertex-weighted context, is:

$$\mathrm{val}_G(T) \leq m(V) \cdot d(V). \tag{2}$$

**Cost.** The original objective introduced by Dasgupta [20] for analyzing hierarchical clustering algorithms introduces the notion of cost.

**Definition 3** (Cost). *The* cost *of a tree $T$ for an instance $G = (V, s)$ where $s(\cdot, \cdot)$ denotes similarity is:* $\mathrm{cost}_G(T) = \sum_{i,j \in V} s(i,j) \cdot |\mathrm{leaves}(T[i \vee j])|$.

The objective is to find a tree of minimum cost. From a complexity point of view, cost is a harder objective to optimize. Charikar and Chatziafratis [12] showed that cost is not constant-factor approximable under the Small Set Expansion hypothesis, and the current best approximations are $O\left(\sqrt{\log n}\right)$ and require solving SDPs.

**Convention.** Throughout the paper we adopt the following convention: $s(\cdot, \cdot)$ will always denote similarities and $d(\cdot, \cdot)$ will always denote distances. Thus, the inputs for the cost and revenue objectives will be instances of the form $(V, s, m)$ and inputs for the value objective will be instances of the form $(V, d, m)$. All the missing proofs can be found in the Supplementary Material.

## 2.2 Notions of fairness

Many definitions have been proposed for fairness in clustering. We consider the setting in which each data point in $V$ has a *color*; the color corresponds to the protected attribute.

*Disparate impact.* This notion is used to capture the fact that decisions (i.e., clusterings) should not be overly favorable to one group versus another. This notion was formalized by Chierichetti et al. [16] for clustering when the protected attribute can take on one of two values, i.e., points have one of two colors. In their setup, the *balance* of a cluster is the ratio of the minimum to the maximum number of points of any color in the cluster. Given a balance requirement $t$, a clustering is fair if and only if each cluster has a balance of at least $t$.

*Bounded representation.* A generalization of disparate impact, bounded representation focuses on mitigating the imbalance of the representation of protected classes (i.e., colors) in clusters and was defined by Ahmadian et al. [3]. Given an over-representation parameter $\alpha$, a cluster is fair if the *fractional representation* of each color in the cluster is at most $\alpha$, and a clustering is fair if each cluster has this property. This was further generalized by Bera et al. [8] and Bercea et al. [9]. They introduce vectors $\vec{\alpha}, \vec{\beta}$ such that for a cluster to be fair, for each color $c_i$, the fractional representation of $c_i$ in the cluster must be between $\beta_i$ and $\alpha_i$. We discuss our results in terms of the over-representation constraint by [3], however many of these results extend to this more general setting given an appropriate fairlet decomposition. An interesting special case of this notion is when there are $c$ total colors and $\alpha = 1/c$. In this case, we require that every color is equally represented in every cluster. We will refer to this as *equal representation*. These notions enjoy the following useful property:

**Definition 4** (Union-closed). *A fairness constraint is* union-closed *if for any pair of fair clusters $A$ and $B$, $A \cup B$ is also fair.*

This property is useful in hierarchical clustering: given a tree $T$ and internal node $u$, if each child cluster of $u$ is fair, then $u$ must also be a fair cluster.

**Definition 5** (Fair hierarchical clustering). *For any fairness constraint, a* hierarchical clustering is fair *if all of its clusters (besides the leaves) are fair.*[1]

Thus, under any union-closed fairness constraint, this definition is equivalent to restricting the bottom-most clustering (besides the leaves) to be fair. Then given an objective (e.g., revenue), the goal is to find a fair hierarchical clustering that optimizes the objective. We focus on the bounded representation fairness notion with $c$ colors and an over-representation cap $\alpha$. However, the main ideas for the revenue and value objectives work under any notion of fairness that is union-closed.

## 3 Fairlet decomposition

**Definition 6** (Fairlet [16]). *A fairlet $Y$ is a fair set of points such that there is no partition of $Y$ into $Y_1$ and $Y_2$ with both $Y_1$ and $Y_2$ being fair.*

In the bounded representation fairness setting, a set of points is fair if at most an $\alpha$ fraction of the points have the same color. We call this an *$\alpha$-capped fairlet*. For $\alpha = 1/t$ with $t$ an integer, the fairlet size will always be at most $2t - 1$. We will refer to the maximum size of a fairlet by $m_f$.

Recall that given a union-closed fairness constraint, if the bottom clustering in the tree is a layer of fairlets (which we call a *fairlet decomposition* of the original dataset) the hierarchical clustering tree is also fair. This observation gives an immediate algorithm for finding fair hierarchical clustering trees in a two-phase manner. (i) Find a fairlet decomposition, i.e., partition the input set $V$ into clusters $Y_1, Y_2, \ldots$ that are all fairlets. (ii) Build a tree on top of all the fairlets. Our goal is to complete both phases in such a way that we optimize the given objective (i.e., revenue or value).

In Section 4, we will see that to optimize for the revenue objective, all we need is a fairlet decomposition with bounded fairlet size. However, the fairlet decomposition required for the value objective is more nuanced. We describe this next.

**Fairlet decomposition for the value objective** For the value objective, we need the total distance between pairs of points inside each fairlet to be small. Formally, suppose $V$ is partitioned into fairlets

$\mathcal{Y} = \{Y_1, Y_2, \ldots\}$ such that $Y_i$ is an $\alpha$-capped fairlet. The cost of this decomposition is defined as:

$$\phi(\mathcal{Y}) = \sum_{Y \in \mathcal{Y}} \sum_{\{u,v\} \subseteq Y} d(u,v). \tag{3}$$

Unfortunately, the problem of finding a fairlet decomposition to minimize $\phi(\cdot)$ does not admit any constant-factor approximation unless P = NP.

**Theorem 7.** *Let $z \geq 3$ be an integer. Then there is no bounded approximation algorithm for finding $(\frac{z}{z+1})$-capped fairlets optimizing $\phi(\mathcal{Y})$, which runs in polynomial time, unless P = NP.*

The proof proceeds by a reduction from the Triangle Partition problem, which asks if a graph $G = (V, E)$ on $3n$ vertices can be partitioned into three element sets, with each set forming a triangle in $G$. Fortunately, for the purpose of optimizing the value objective, it is not necessary to find an approximate decomposition.

## 4 Optimizing revenue with fairness

This section considers the revenue objective. We will obtain an approximation algorithm for this objective in three steps: (i) obtain a fairlet decomposition such that the maximum fairlet size in the decomposition is small, (ii) show that any $\beta$-approximation algorithm to (1) (i.e., any algorithm that achieves a $\beta$-factor approximation of (1) for some given $\beta$) plus this fairlet decomposition can be used to obtain a (roughly) $\beta$-approximation for fair hierarchical clustering under the revenue objective, and (iii) use average-linkage, which is known to be a $1/3$-approximation to (1). (We note that the recent work [2, 5] on improved approximation algorithms compare to a bound on the optimal solution that differs from (1) and therefore do not fit into our framework.)

First, we address step (ii). Due to space, this proof can be found in Appendix B. Note that Theorem 8 extends to the fairness constraint defined by [8, 9]'s provided a fairlet decomposition in this setting.

**Theorem 8.** *Given an algorithm that obtains a $\beta$-approximation to (1) where $\beta \leq 1$, and a fairlet decomposition with maximum fairlet size $m_f$, there is a $\beta\left(1 - \frac{2m_f}{n}\right)$-approximation for fair hierarchical clustering under the revenue objective.*

Prior work showed that average-linkage is a $1/3$-approximation to (1) in the vertex-unweighted case; this proof can be easily modified to show that it is still $1/3$-approximation even with vertex weights. This accounts for step (iii) in our process.

Combined with the fairlet decomposition methods for the two-color case [16], which has $m_f = b + r$ for $b$ blue vertices and $r$ red vertices, and for multi-color case (Supplementary Material), which has $m_f \leq 2t - 1$, to address step (i), we have the following.

**Corollary 9.** *There is polynomial time algorithm that constructs a fair tree that is a $\frac{1}{3}\left(1 - \frac{2m_f}{n}\right)$-approximation for revenue objective, where $m_f$ is the maximum size of fairlets.*

## 5 Optimizing value with fairness

In this section we consider the value objective. As in the revenue objective, we prove that we can reduce fair hierarchical clustering to the problem of finding a good fairlet decomposition for the proposed fairlet objective (3), and then use any approximation algorithm for weighted hierarchical clustering with the decomposition as the input. Again, our result applies to [8, 9]'s fairness constraint if we are given an appropriate fairness decomposition.

**Theorem 10.** *Given an algorithm that gives a $\beta$-approximation to (2) where $\beta \leq 1$, and a fairlet decomposition $\mathcal{Y}$ such that $\phi(\mathcal{Y}) \leq \epsilon \cdot d(V)$, there is a $\beta(1 - \epsilon)$ approximation for (2).*

We complement this result with an algorithm that finds a good fairlet decomposition in polynomial time under the bounded representation fairness constraint with cap $\alpha$.

Let $R_1, \ldots, R_c$ be the $c$ colors and let $\mathcal{Y} = \{Y_1, Y_2 \ldots\}$ be the fairlet decomposition. Let $n_i$ be the number of points colored $R_i$ in $V$. Let $r_{i,k}$ denote the number of points colored $R_i$ in the $k$th fairlet.

**Theorem 11.** *There exists a local search algorithm that finds a fairlet decomposition $\mathcal{Y}$ with $\phi(\mathcal{Y}) \leq (1 + \epsilon) \max_{i,k} \frac{r_{i,k}}{n_i} d(V)$ in time $\tilde{O}(n^3/\epsilon)$.*

We can now use the fact that both average-linkage and the $\frac{\epsilon}{n}$-locally-densest cut algorithm give a $\frac{2}{3}$- and $(\frac{2}{3} - \epsilon)$-approximation respectively for vertex-weighted hierarchical clustering under the value objective. Finally, recall that fairlets are intended to be minimal, and their size depends only on the parameter $\alpha$, and not on the size of the original input. Therefore, as long as the number of points of each color increases as input size, $n$, grows, the ratio $r_{i,k}/n_i$ goes to 0. These results, combined with Theorem 10 and Theorem 11, yield Corollary 12.

**Corollary 12.** *Given bounded size fairlets, the fairlet decomposition computed by local search combined with average-linkage constructs a fair hierarchical clustering that is a $\frac{2}{3}(1 - o(1))$-approximation for the value objective. For the $\frac{\epsilon}{n}$-locally-densest cut algorithm in [19], we get a polynomial time algorithm for fair hierarchical clustering that is a $(\frac{2}{3} - \epsilon)(1 - o(1))$-approximation under the value objective for any $\epsilon > 0$.*

Given at most a small fraction of every color is in any cluster, Corollary 12 states that we can extend the state-of-the-art results for value to the $\alpha$-capped, multi-colored constraint. Note that the preconditions will always be satisfied and the extension will hold in the two-color fairness setting or in the multi-colored equal representation fairness setting.

**Fairlet decompositions via local search**  In this section, we give a local search algorithm to construct a fairlet decomposition, which proves Theorem 11. This is inspired by the $\epsilon$-densest cut algorithm of [19]. To start, recall that for a pair of sets $A$ and $B$ we denote by $d(A, B)$ the sum of interpoint distances, $d(A, B) = \sum_{u \in A, v \in B} d(u, v)$. A fairlet decomposition is a partition of the input $\{Y_1, Y_2, \ldots\}$ such that each color composes at most an $\alpha$ fraction of each $Y_i$.

We start by finding an arbitrary $\alpha$-capped fairlet decomposition. For two colors with $\alpha = r/(b + r)$, we use the fairlet decomposition introduced by Chierichetti et al. [16]. For multiple colors with $\alpha = 1/t$, we defer to Lemma 24 in Appendix C.2. Our algorithm will then recursively subdivide the cluster of all data to construct a hierarchy by finding cuts. To search for a cut, we will use a *swap* method.

**Definition 13** (Local optimality). *Consider any fairlet decomposition $\mathcal{Y} = \{Y_1, Y_2, \ldots\}$ and $\epsilon > 0$. Define a swap of $u \in Y_i$ and $v \in Y_j$ for $j \neq i$ as updating $Y_i$ to be $(Y_i \setminus \{u\}) \cup \{v\}$ and $Y_j$ to be $(Y_j \setminus \{v\}) \cup \{u\}$. We say $\mathcal{Y}$ is $\epsilon$-locally-optimal if any swap with $u, v$ of the same color reduces the objective value by less than a $(1 + \epsilon)$ factor.*

The algorithm constructs a $(\epsilon/n)$-locally optimal algorithm for fairlet decomposition, which runs in $\tilde{O}(n^3/\epsilon)$ time. Consider any given instance $(V, d)$. Let $d_{\max}$ denote the maximum distance, $m_f$ denote the maximum fairlet size, and $\Delta = d_{\max} \cdot \frac{m_f}{n}$. The algorithm begins with an arbitrary decomposition. Then it swaps pairs of monochromatic points until it terminates with a locally optimal solution. By construction we have the following.

**Claim 14.** *Algorithm 1 finds a valid fairlet decomposition.*

We prove two things: Algorithm 1 optimizes the objective (3), and has a small running time. The following lemma gives an upper bound on $\mathcal{Y}$'s performance for (3) found by Algorithm 1.

**Lemma 15.** *The fairlet decomposition $\mathcal{Y}$ computed by Algorithm 1 has an objective value for (3) of at most $(1 + \epsilon) \max_{i,k} \frac{r_{i,k}}{n_i} d(V)$.*

Finally we bound the running time. The algorithm has much better performance in practice than its worst-case analysis would indicate. We will show this later in Section 7.

**Lemma 16.** *The running time for Algorithm 1 is $\tilde{O}(n^3/\epsilon)$.*

Together, Lemma 15, Lemma 16, and Claim 14 prove Theorem 11. This establishes that there is a local search algorithm that can construct a good fairlet decomposition.

---

**Algorithm 1** Algorithm for $(\epsilon/n)$-locally-optimal fairlet decomposition.

---

**Input:** A set $V$ with distance function $d \geq 0$, parameter $\alpha$, small constant $\epsilon \in [0,1]$.
**Output:** An $\alpha$-capped fairlet decomposition $\mathcal{Y}$.

1: Find $d_{\max}$, $\Delta \leftarrow \frac{m_f}{n} d_{\max}$.
2: Arbitrarily find an $\alpha$-capped fairlet decomposition $\{Y_1, Y_2, \ldots\}$ such that each partition has at most an $\alpha$ fraction of any color. {See [16] or Appendix C.2}
3: **while** $\exists u \in Y_i, v \in Y_j, i \neq j$ of the same color, such that for the decomposition $\mathcal{Y}'$ after swapping $u, v$, $\frac{\sum_{Y_k \in \mathcal{Y}} d(Y_k)}{\sum_{Y_k \in \mathcal{Y}'} d(Y_k)} \geq (1 + \epsilon/n)$ **and** $\sum_{Y_k \in \mathcal{Y}} d(Y_k) > \Delta$ **do**
4:     Swap $u$ and $v$ by setting $Y_i \leftarrow (Y_i \setminus \{u\}) \cup \{v\}$ and $Y_j \leftarrow (Y_j \setminus \{v\}) \cup \{u\}$.
5: **end while**

---

Table 1: Dataset description. Here $(b, r)$ denotes the balance of the dataset.

| Name | Sample size | # features | Protected feature | Color (blue, red) | $(b, r)$ |
|---|---|---|---|---|---|
| CENSUSGENDER | 30162 | 6 | gender | (female, male) | $(1, 3)$ |
| CENSUSRACE | 30162 | 6 | race | (non-white, white) | $(1, 7)$ |
| BANKMARRIAGE | 45211 | 7 | marital status | (not married, married) | $(1, 2)$ |
| BANKAGE | 45211 | 7 | age | $(< 40, \geq 40)$ | $(2, 3)$ |

Table 2: Impact of Algorithm 1 on ratio$_{\text{value}}$ in percentage (mean $\pm$ std. dev).

| Samples | 400 | 800 | 1600 | 3200 | 6400 | 12800 |
|---|---|---|---|---|---|---|
| CENSUSGENDER, initial | $88.17 \pm 0.76$ | $88.39 \pm 0.21$ | $88.27 \pm 0.40$ | $88.12 \pm 0.26$ | $88.00 \pm 0.10$ | $88.04 \pm 0.13$ |
| final | $99.01 \pm 0.60$ | $99.09 \pm 0.58$ | $99.55 \pm 0.26$ | $99.64 \pm 0.13$ | $99.20 \pm 0.38$ | $99.44 \pm 0.23$ |
| CENSUSRACE, initial | $84.49 \pm 0.66$ | $85.01 \pm 0.31$ | $85.00 \pm 0.42$ | $84.88 \pm 0.43$ | $84.84 \pm 0.16$ | $84.89 \pm 0.20$ |
| final | $99.50 \pm 0.20$ | $99.89 \pm 0.32$ | $100.0 \pm 0.21$ | $99.98 \pm 0.21$ | $99.98 \pm 0.11$ | $99.93 \pm 0.31$ |
| BANKMARRIAGE, initial | $92.47 \pm 0.54$ | $92.58 \pm 0.30$ | $92.42 \pm 0.30$ | $92.53 \pm 0.14$ | $92.59 \pm 0.14$ | $92.75 \pm 0.04$ |
| final | $99.18 \pm 0.22$ | $99.28 \pm 0.33$ | $99.59 \pm 0.14$ | $99.51 \pm 0.17$ | $99.46 \pm 0.10$ | $99.50 \pm 0.05$ |
| BANKAGE, initial | $93.70 \pm 0.56$ | $93.35 \pm 0.41$ | $92.95 \pm 0.25$ | $93.28 \pm 0.13$ | $93.36 \pm 0.12$ | $93.33 \pm 0.12$ |
| final | $99.40 \pm 0.28$ | $99.40 \pm 0.51$ | $99.61 \pm 0.13$ | $99.64 \pm 0.07$ | $99.65 \pm 0.08$ | $99.59 \pm 0.06$ |

## 6 Optimizing cost with fairness

This section considers the cost objective of [20]. Even without our fairness constraint, the difficulty of approximating cost is clear in its approximation hardness and the fact that all known solutions require an LP or SDP solver. We obtain the result in Theorem 17; extending this result to other fairness constraints, improving its bound, or even making the algorithm practical, are open questions.

**Theorem 17.** *Consider the two-color case. Given a $\beta$-approximation for cost and a $\gamma_t$-approximation for minimum weighted bisection [2] on input of size $t$, then for parameters $t$ and $\ell$ such that $n \geq t\ell$ and $n > \ell + 108t^2/\ell^2$, there is a fair $O\left(\frac{n}{t} + t\ell + \frac{n\ell\gamma_t}{t} + \frac{nt\gamma_t}{\ell^2}\right)$ $\beta$-approximation for $\mathrm{cost}(T^*_{\text{unfair}})$.*

With proper parameterization, we achieve an $O\left(n^{5/6} \log^{5/4} n\right)$-approximation. We defer our algorithm description, pseudocode, and proofs to the Supplementary Material. While our algorithm is not simple, it is an important (and non-obvious) step to show the existence of an approximation, which we hope will spur future work in this area.

## 7 Experiments

This section validates our algorithms from Sections 4 and 5 empirically. We adopt the disparate impact fairness constraint [16]; thus each point is either blue or red. In particular, we would like to:

- Show that running the standard average-linkage algorithm results in highly unfair solutions.
- Demonstrate that demanding fairness in hierarchical clustering incurs only a small loss in the hierarchical clustering objective.
- Show that our algorithms, including fairlet decomposition, are practical on real data.

In Appendix G we consider multiple colors and the same trends as the two color case occur.

Table 3: Impact of Algorithm 1 on $\text{ratio}_{\text{fairlets}}$.

| Samples | 100 | 200 | 400 | 800 | 1600 | 3200 | 6400 | 12800 |
|---|---|---|---|---|---|---|---|---|
| CENSUSGENDER, initial | 2.5e-2 | 1.2e-2 | 6.2e-3 | 3.0e-3 | 1.5e-3 | 7.5e-4 | 3.8e-4 | 1.9e-4 |
| final | 4.9e-3 | 1.4e-3 | 6.9e-4 | 2.5e-4 | 8.5e-5 | 3.6e-5 | 1.8e-5 | 8.0e-6 |
| CENSUSRACE, initial | 6.6e-2 | 3.4e-2 | 1.7e-2 | 8.4e-3 | 4.2e-3 | 2.1e-3 | 1.1e-3 | 5.3e-4 |
| final | 2.5e-2 | 1.2e-2 | 6.2e-3 | 3.0e-3 | 1.5e-3 | 7.5e-4 | 3.8e-4 | 1.9e-5 |
| BANKMARRIAGE, initial | 1.7e-2 | 8.2e-3 | 4.0e-3 | 2.0e-3 | 1.0e-3 | 5.0e-4 | 2.5e-4 | 1.3e-4 |
| final | 5.9e-3 | 2.1e-3 | 9.3e-4 | 4.1e-4 | 1.3e-4 | 7.1e-5 | 3.3e-5 | 1.4e-5 |
| BANKAGE, initial | 1.3e-2 | 7.4e-3 | 3.5e-3 | 1.9e-3 | 9.3e-4 | 4.7e-4 | 2.3e-4 | 1.2e-4 |
| final | 5.0e-3 | 2.2e-3 | 7.0e-4 | 3.7e-4 | 1.3e-4 | 5.7e-5 | 3.0e-5 | 1.4e-5 |

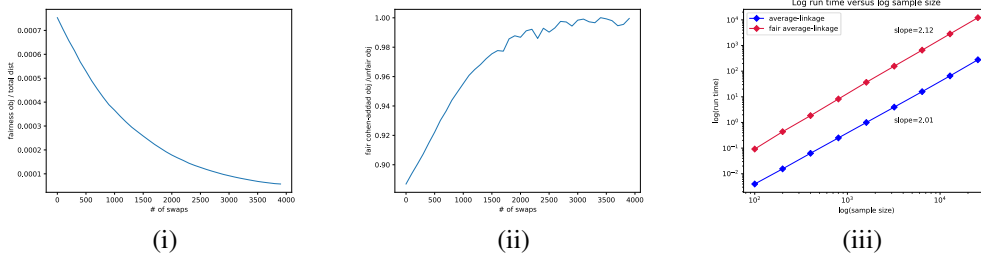

| (i) | (ii) | (iii) |

Figure 1: (i) $\text{ratio}_{\text{fairlets}}$, every 100 swaps. (ii) $\text{ratio}_{\text{value}}$, every 100 swaps. (iii) CENSUSGENDER: running time vs sample size on a log-log scale.

**Datasets.** We use two datasets from the UCI data repository.[3] In each dataset, we use features with numerical values and leave out samples with empty entries. For value, we use the Euclidean distance as the dissimilarity measure. For revenue, we set the similarity to be $s(i,j) = \frac{1}{1+d(i,j)}$ where $d(i,j)$ is the Euclidean distance. We pick two different protected features for both datasets, resulting in four datasets in total (See Table 1 for details).

- *Census* dataset: We choose *gender* and *race* to be the protected feature and call the resulting datasets CENSUSGENDER and CENSUSRACE.
- *Bank* dataset: We choose *marital status* and *age* to be the protected features and call the resulting datasets BANKMARRIAGE and BANKAGE.

In this section, unless otherwise specified, we report results only for the value objective. Results for the revenue objective are qualitatively similar and are omitted here. We do not evaluate our algorithm for the cost objective since it is currently only of theoretical interest.

We sub-sample points of two colors from the original data set proportionally, while approximately retaining the original color balance. The sample sizes used are $100 \times 2^i, i = 0, \ldots, 8$. On each, we do 5 experiments and report the average results. We set $\epsilon$ in Algorithm 1 to 0.1 in all of the experiments.

**Implementation.** The code is available in the Supplementary Material. In the experiments, we use Algorithm 1 for the fairlet decomposition phase, where the fairlet decomposition is initialized by randomly assigning red and blue points to each fairlet. We apply the average-linkage algorithm to create a tree on the fairlets. We further use average-linkage to create subtrees inside of each fairlet.

The algorithm selects a *random* pair of blue or red points in different fairlets to swap, and checks if the swap sufficiently improves the objective. We do not run the algorithm until all the pairs are checked, rather the algorithm stops if it has made a $2n$ failed attempts to swap a random pair. As we observe empirically, this does not have material effect on the quality of the overall solution.

**Metrics.** We present results for value here, the results for revenue are qualitatively similar. In our experiments, we track the following quantities. Let $G$ be the given input instance and let $T$ be the output of our fair hierarchical clustering algorithm. We consider the following ratio $\text{ratio}_{\text{value}} = \frac{\text{value}_G(T)}{\text{value}_G(T')}$, where $T'$ is the tree obtained by the standard average-linkage algorithm. We consider the fairlet objective function where $\mathcal{Y}$ is a fairlet decomposition. Let $\text{ratio}_{\text{fairlets}} = \frac{\phi(\mathcal{Y})}{d(V)}$.

Table 4: Clustering on fairlets found by local search vs. upper bound, at size 1600 (mean $\pm$ std. dev).

| Dataset | CENSUSGENDER | CENSUSRACE | BANKMARRIAGE | BANKAGE |
|---|---|---|---|---|
| Revenue vs. upper bound | $81.89 \pm 0.40$ | $81.75 \pm 0.83$ | $61.53 \pm 0.37$ | $61.66 \pm 0.66$ |
| Value vs. upper bound | $84.31 \pm 0.15$ | $84.52 \pm 0.22$ | $89.17 \pm 0.29$ | $88.81 \pm 0.18$ |

**Results.** Average-linkage algorithm always constructs unfair trees. For each of the datasets, the algorithm results in monochromatic clusters at some level, strengthening the case for fair algorithms.

In Table 2, we show for each dataset the $\text{ratio}_{\text{value}}$ both at the time of initialization (Initial) and after using the local search algorithm (Final). We see the change in the ratio as the local search algorithm performs swaps. Fairness leads to almost no degradation in the objective value as the swaps increase. Table 3 shows the $\text{ratio}_{\text{fairlets}}$ between the initial initialization and the final output fairlets. As we see, Algorithm 1 significantly improves the fairness of the initial random fairlet decomposition. The more the locally-optimal algorithm improves the objective value of (3), the better the tree's performance based on the fairlets. Figures 1(i) and 1(ii) show $\text{ratio}_{\text{value}}$ and $\text{ratio}_{\text{fairlets}}$ for every 100 swaps in the execution of Algorithm 1 on a subsample of size 3200 from Census data set. The plots show that as the fairlet objective value decreases, the value objective of the resulting fair tree increases. Such correlation are found on subsamples of all sizes.

Now we compare the objective value of the algorithm with the upper bound on the optimum. We report the results for both the revenue and value objectives, using fairlets obtained by local search, in Table 4. On all datasets, we obtain ratios significantly better than the theoretical worst case guarantee. In Figure 1(iii), we show the average running time on Census data for both the original average-linkage and the fair average-linkage algorithms. As the sample size grows, the running time scales almost as well as current implementations of average-linkage algorithm. Thus with a modest increase in time, we can obtain a fair hierarchical clustering under the value objective.

# 8 Conclusions

In this paper we extended the notion of fairness to the classical problem of hierarchical clustering under three different objectives (revenue, value, and cost). Our results show that revenue and value are easy to optimize with fairness; while optimizing cost appears to be more challenging.

Our work raises several questions and research directions. Can the approximations be improved? Can we find better upper and lower bounds for fair cost? Are there other important fairness criteria?

## Broader Impact

Our work builds upon a long line of work of fairness in machine learning. See the excellent books by Kearns and Roth [27], and Barocas et al. [7] for a rich introduction to the field.

Our aim in this work is algorithmic in nature, finding near-optimal hierarchical clustering algorithms that attain certain fairness guarantees. Since these methods are common unsupervised learning primitives, it is important to develop tools for practitioners to use. At the same time we remark that just because an algorithm is proven to be "fair" under some definition, does not mean it can be applied blindly.

As is now well known, [29], different fairness notions can be incompatible with each other. Moreover, fairness in machine learning is necessarily problem specific, and depends on the goals and the values of the person invoking the algorithm. While these facts are well established in the research community, they are far from common knowledge outside of it. Thus work on algorithmic notions of fairness runs the risk of someone treating the results as a silver bullet, and eschewing the deeper analysis that is necessary in any real world application.

## Funding Sources

B. Moseley and Y. Wang were supported in part by a Google Research Award, an Infor Research Award, a Carnegie Bosch Junior Faculty Chair and NSF grants CCF-1824303, CCF-1845146, CCF-1733873 and CMMI-1938909. B. Moseley additionally is a part time employee of Relational-AI. M. Knittel was supported in part by NSF BIGDATA grant IIS-1546108, and NSF SPX grant CCF-1822738 and some of the work was conducted while she was visiting Google.

## Footnotes

[1] According to the definition, a hierarchical clustering tree might be fair even if every layer (apart from the root) is an unfair clustering. For example, consider a tree that splits off one singleton at its root. Every layer in the tree apart from the root will contain this singleton and thus is an unfair clustering. An alternative way of defining a fair tree is to enforce that the tree has to contain a layer of fairlets of some small size. The results of this paper extend to either definition.

[2]The minimum weighted bisection problem is to find a partition of nodes into two equal-sized subsets so that the sum of the weights of the edges crossing the partition is minimized.

[3] archive.ics.uci.edu/ml/index.php. Census: archive.ics.uci.edu/ml/datasets/census+income, Bank: archive.ics.uci.edu/ml/datasets/Bank+Marketing

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
