[Supplementary Material]

# Appendix

## A   Approximation algorithms for weighted hierarchical clustering

In this section we first prove that running constant-approximation algorithms on fairlets gives good solutions for value objective, and then give constant approximation algorithms for both revenue and value in weighted hierarchical clustering problem, as is mentioned in Corollary 9 and 12. That is, a weighted version of average-linkage, for both weighted revenue and value objective, and weighted $(\epsilon/n)$-locally densest cut algorithm, which works for weighted value objective. Both proofs are easily adapted from previous proofs in [19] and [33].

### A.1   Running constant-approximation algorithms on fairlets

In this section, we prove Theorem 10, which says if we run any $\beta$-approximation algorithm for the upper bound on weighted value on the fairlet decomposition, we get a fair tree with minimal loss in approximation ratio. For the remainder of this section, fix any hierarchical clustering algorithm $A$ that is guaranteed on any *weighted* input $(V, d, m)$ to construct a hierarchical clustering with objective value at least $\beta m(V) d(V)$ for the value objective on a weighted input. Recall that we extended the value objective to a weighted variant in the Preliminaries Section and $m(V) = \sum_{u \in V} m_u$. Our aim is to show that we can combine $A$ with the fairlet decomposition $\mathcal{Y}$ introduced in the prior section to get a fair hierarchical clustering that is a $\beta(1 - \epsilon)$-approximation for the value objective, if $\phi(\mathcal{Y}) \le \epsilon d(V)$.

In the following definition, we transform the point set to a new set of points that are weighted. We will analyze $A$ on this new set of points. We then show how we can relate this to the objective value of the optimal tree on the original set of points.

**Definition 18.** *Let $\mathcal{Y} = \{Y_1, Y_2, \ldots\}$ be the fairlet decomposition for $V$ that is produced by the local search algorithm. Define $V(\mathcal{Y})$ as follows:*

- *Each set $Y_i$ has a corresponding point $a_i$ in $V(\mathcal{Y})$.*

- *The weight $m_i$ of $a_i$ is set to be $|Y_i|$.*

- *For each partitions $Y_i, Y_j$, where $i \ne j$ and $Y_i, Y_j \in \mathcal{Y}$, $d(a_i, a_j) = d(Y_i, Y_j)$.*

We begin by observing the objective value that $A$ receives on the instance $V(\mathcal{Y})$ is large compared to the weights in the original instance.

**Theorem 19.** *On the instance $V(\mathcal{Y})$ the algorithm $A$ has a total weighted objective of $\beta(1 - \epsilon) \cdot nd(V)$.*

*Proof.* Notice that $m(V(\mathcal{Y})) = |V| = n$. Consider the total sum of all the distances in $V(\mathcal{Y})$. This is $\sum_{a_i, a_j \in V(\mathcal{Y})} d(a_i, a_j) = \sum_{Y_i, Y_j \in \mathcal{Y}} d(Y_i, Y_j) = d(V) - \phi(\mathcal{Y})$. The upper bound on the optimal solution is $(\sum_{Y_i \in \mathcal{Y}} m_i)(d(V) - \phi(\mathcal{Y}) = n(d(V) - \phi(\mathcal{Y}))$. Since $\phi(\mathcal{Y}) \le \epsilon d(V)$, this upper bound is at least $(1 - \epsilon)nd(V)$. Theorem 10 follows from the fact that the algorithm $A$ archives a weighted revenue at least a $\beta$ factor of the total weighted distances.  $\square$

### A.2   Weighted hierarchical clustering: Constant-factor approximation

For weighted hierarchical clustering with positive integral weights, we define the weighted average-linkage algorithm for input $(V, d, m)$ and $(V, s, m)$. Define the *average distance* to be $Avg(A, B) = \frac{d(A,B)}{m(A)m(B)}$ for dissimilarity-based input, and $Avg(A, B) = \frac{s(A,B)}{m(A)m(B)}$ for similarity-based input. In each iteration, weighted average-linkage seeks to merge the clusters which minimizes this value, if dissimilarity-based, and maximizes this value, if similarity-based.

**Lemma 20.** *Weighted average-linkage is a $\frac{2}{3}(resp., \frac{1}{3})$ approximation for the upper bound on weighted value (resp., revenue) objective with positive, integral weights.*

*Proof.* We prove it for weighted value first. This is directly implied by the fact that average-linkage is $\frac{2}{3}$ approximation for unweighted value objective, as is proved in [19]. We have already seen in the

last subsection that a unweighted input $V$ can be converted into weighted input $V(\mathcal{Y})$. Vice versa, we can construct a weighted input $(V, d, m)$ into unweighted input with same upper bound for value objective.

In weighted hierarchical clustering we treat each point $p$ with integral weights as $m(p)$ duplicates of points with distance 0 among themselves, let's call this set $S(p)$. For two weighted points $(p, m(p))$ and $(q, m(q))$, if $i \in S(p), j \in S(q)$, let $d(i, j) = \frac{d(p,q)}{m(p)m(q)}$. This *unweighted* instance, composed of many duplicates, has the same upper bound as the weighted instance. Notice that running average-linkage on the unweighted instance will always choose to put all the duplicates $S(p)$ together first for each $p$, and then do hierarchical clustering on top of the duplicates. Thus running average-linkage on the unweighted input gives a valid hierarchical clustering tree for weighted input. Since unweighted value upper bound equals weighted value upper bound, the approximation ratio is the same.

Now we prove it for weighted revenue. In [33], average-linkage being $\frac{1}{3}$ approximation for un-weighted revenue is proved by the following. Given any clustering $\mathcal{C}$, if average-linkage chooses to merge $A$ and $B$ in $\mathcal{C}$, we define a local revenue for this merge:

$$\text{merge-rev}(A, B) = \sum_{C \in \mathcal{C} \setminus \{A,B\}} |C||A||B|Avg(A, B).$$

And correspondingly, a local cost:

$$\text{merge-cost}(A, B) = \sum_{C \in \mathcal{C} \setminus \{A,B\}} (|B||A||C|Avg(A, C) + |A||B||C|Avg(B, C)).$$

Summing up the local revenue and cost over all merges gives the upper bound. [33] used the property of average-linkage to prove that at every merge, $\text{merge-cost}(A, B) \leq 2\text{merge-rev}(A, B)$, which guarantees the total revenue, which is the summation of $\text{merge-rev}(A, B)$ over all merges, is at least $\frac{1}{3}$ of the upper bound. For the weighted case, we define

$$\text{merge-rev}(A, B) = \sum_{C \in \mathcal{C} \setminus \{A,B\}} m(C)m(A)m(B)Avg(A, B).$$

And

$$\text{merge-cost}(A, B) \sum_{C \in \mathcal{C} \setminus \{A,B\}} (m(B)m(A)m(C)Avg(A, C) + m(A)m(B)m(C)Avg(B, C)).$$

And the rest of the proof works in the same way as in [33], proving weighted average-linkage to be $\frac{1}{3}$ for weighted revenue. $\square$

Next we define the weighted $(\epsilon/n)$-locally-densest cut algorithm. The original algorithm, introduced in [19], defines a cut to be $\frac{d(A,B)}{|A||B|}$. It starts with the original set as one cluster, at every step, it seeks the partition of the current set that locally maximizes this value, and thus constructing a tree from top to bottom. For the weighted input $(V, d, m)$, we define the cut to be $\frac{d(A,B)}{m(A)m(B)}$, and let $n = m(V)$. For more description of the algorithm, see Algorithm 4 in Section 6.2 in [19].

**Lemma 21.** *Weighted $(\epsilon/n)$-locally-densest cut algorithm is a $\frac{2}{3} - \epsilon$ approximation for weighted value objective.*

*Proof.* Just as in the average-linkage proof, we convert each weighted point $p$ into a set $S$ of $m(p)$ duplicates of $p$. Notice that the converted unweighted hierarchical clustering input has the same upper bound as the weighted hierarchical clustering input, and the $\epsilon/n$-locally-densest cut algorithm moves all the duplicate sets $S$ around in the unweighted input, instead of single points as in the original algorithm in [19].

Focus on a split of cluster $A \cup B$ into $(A, B)$. Let $S$ be a duplicate set. $\forall S \subseteq A$, where $S$ is a set of duplicates, we must have

$$(1 + \frac{\epsilon}{n})\frac{d(A, B)}{|A||B|} \geq \frac{d(A \setminus S, B \cup S)}{(|A| - |S|)(|B| + |S|)}.$$

Pick up a point $q \in S$,

$$(1 + \frac{\epsilon}{n})d(A,B)|S|(|A|-1)(|B|+1)$$

$$= (1 + \frac{\epsilon}{n})d(A,B)(|A||B| + |A| - |B| - 1)|S|$$

$$= (1 + \frac{\epsilon}{n})d(A,B)(|A||B| + |A||S| - |B||S| - |S|) + (1 + \frac{\epsilon}{n})d(A,B)(|A||B|)(|S|-1)$$

$$\geq (1 + \frac{\epsilon}{n})d(A,B)(|A| - |S|)(|B| + |S|) + d(A,B)|A||B|(|S|-1)$$

$$\geq |A||B|d(A \setminus S, B \cup S) + d(A,B)|A||B|(|S|-1)$$

$$= |A||B|(d(A,B) + |S|d(q,A) - |S|d(q,B)) + |A||B|(|S|-1)d(A,B)$$

$$= |A||B||S|(d(A,B) + d(q,A) - d(q,B)).$$

Rearrange the terms and we get the following inequality holds for any point $q \in A$:

$$\left(1 + \frac{\epsilon}{n}\right)\frac{d(A,B)}{|A||B|} \geq \frac{d(A,B) + d(q,A) - d(q,B)}{(|A|-1)(|B|+1)}.$$

The rest of the proof goes exactly the same as the proof in [19, Theorem 6.5]. $\square$

## B  Proof of Theorem 8

*Proof.* Let $\mathcal{A}$ be the $\beta$-approximation algorithm to (1). For a given instance $G = (V, s)$, let $\mathcal{Y} = \{Y_1, Y_2, \ldots\}$ be a fairlet decomposition of $V$; let $m_f = \max_{Y \in \mathcal{Y}} |Y|$. Recall that $n = |V|$.

We use $\mathcal{Y}$ to create a weighted instance $G_\mathcal{Y} = (\mathcal{Y}, s_\mathcal{Y}, m_\mathcal{Y})$. For $Y, Y' \in \mathcal{Y}$, we define $s(Y, Y') = \sum_{i \in Y, j \in Y'} s(i, j)$ and we define $m_\mathcal{Y}(Y) = |Y|$.

We run $\mathcal{A}$ on $G_\mathcal{Y}$ and let $T_\mathcal{Y}$ be the hierarchical clustering obtained by $\mathcal{A}$. To extend this to a tree $T$ on $V$, we simply place all the points in each fairlet as leaves under the corresponding vertex in $T_\mathcal{Y}$.

We argue that $\mathrm{rev}_G(T) \geq \beta \left(1 - \frac{2m_f}{n}\right)(n-2)s(V)$.

Since $\mathcal{A}$ obtains a $\beta$-approximation to hierarchical clustering on $G_\mathcal{Y}$, we have $\mathrm{rev}_{G_\mathcal{Y}}(T_\mathcal{Y}) \geq \beta \cdot \sum_{Y, Y' \in \mathcal{Y}} s(Y, Y')(n - m(Y) - m(Y'))$.

Notice the fact that, for any pair of points $u, v$ in the same fairlet $Y \in \mathcal{Y}$, the revenue they get in the tree $T$ is $(n - m(Y))s(u, v)$. Then using $rev_G(T) = \sum_{Y \in \mathcal{Y}}(n - m(Y))s(Y) + \mathrm{rev}(T_\mathcal{Y})$,

$$\mathrm{rev}_G(T) \geq \sum_{Y \in \mathcal{Y}} \beta(n - m(Y))s(Y) + \beta \sum_{Y, Y' \in \mathcal{Y}} s(Y, Y')(n - m(Y) - m(Y'))$$

$$\geq \beta(n - 2m_f)\left(\sum_{Y \in \mathcal{Y}} s(Y) + \sum_{Y, Y' \in \mathcal{Y}} s(Y, Y')\right) \geq \beta\left(1 - \frac{2m_f}{n}\right)(n-2)s(V).$$

Thus the resulting tree $T$ is a $\beta\left(1 - \frac{2m_f}{n}\right)$-approximation of the upper bound. $\square$

## C  Proofs for $(\epsilon/n)$-locally-optimal local search algorithm

In this section, we prove that Algorithm 1 gives a good fairlet decomposition for the fairlet decomposition objective 3, and that it has polynomial run time.

### C.1  Proof for a simplified version of Lemma 15

In Subsection C.2, we will prove Lemma 15. For now, we will consider a simpler version of Lemma 15 in the context of [16]'s disparate impact problem, where we have red and blue points and strive to preserve their ratios in all clusters. Chierichetti et al. [16] provided a valid fairlet decomposition in this context, where each fairlet has at most $b$ blue points and $r$ red points. Before going deeper into the analysis, we state the following useful proposition.

**Proposition 22.** *Let $r_t = |red(V)|$ be the total number of red points and $b_t = |blue(V)|$ the number of blue points. We have that, $\max\{\frac{r}{r_t}, \frac{b}{b_t}\} \le \frac{2(b+r)}{n}$.*

*Proof.* Recall that $balance(V) = \frac{b_t}{r_t} \ge \frac{b}{r}$, and wlog $b_t \le r_t$. Since the fractions are positive and $\frac{b_t}{r_t} \ge \frac{b}{r}$ we know that $\frac{b_t}{b_t + r_t} \ge \frac{b}{b+r}$. Since $b_t + r_t = n$ we conclude that $b_t \ge \frac{b}{b+r}n$. Similarly, we conclude that $\frac{r_t}{b_t + r_t} \le \frac{r}{b+r}$. Therefore $r_t \le \frac{r}{b+r}n$.

Thus, $\frac{r}{r_t} \ge \frac{b+r}{n} \ge \frac{b}{b_t}$. However, since $b_t \le r_t$ and $b_t + r_t = n$, $r_t \ge \frac{1}{2}n$, $\frac{r}{r_t} \le \frac{2r}{n} \le \frac{2(b+r)}{n}$. $\square$

Using this, we can define and prove the following lemma, which is a simplified version of Lemma 15.

**Lemma 23.** *The fairlet decomposition $\mathcal{Y}$ computed by Algorithm 1 has an objective value for (3) of at most $(1 + \epsilon)\frac{2(b+r)}{n}d(V)$.*

*Proof.* Let $Y : V \mapsto \mathcal{Y}$ denote a mapping from a point in $V$ to the fairlet it belongs to. Let $d_R(X) = \sum_{u \in red(X)} d(u, X)$, and $d_B(X) = \sum_{v \in blue(X)} d(v, X)$. Naturally, $d_R(X) + d_B(X) = 2d(X)$ for any set $X$. For a fairlet $Y_i \in \mathcal{Y}$, let $r_i$ and $b_i$ denote the number of red and blue points in $Y_i$.

We first bound the total number of intra-fairlet pairs. Let $x_i = |Y_i|$, we know that $0 \le x_i \le b + r$ and $\sum_i x_i = n$. The number of intra-fairlet pairs is at most $\sum_i x_i^2 \le \sum_i (b+r)x_i = (b+r)n$.

The **While** loop can end in two cases: 1) if $\mathcal{Y}$ is $(\epsilon/n)$-locally-optimal; 2) if $\sum_{Y_k \in \mathcal{Y}} d(Y_k) \le \Delta$. Case 2 immediately implies the lemma, thus we focus on case 1. By definition of the algorithm, we know that for any pair $u \in Y(u)$ and $v \in Y(v)$ where $u, v$ have the same color and $Y(u) \ne Y(v)$ the swap does not increase objective value by a large amount. (The same trivially holds if the pair are in the same cluster.)

$$\sum_{Y_k} d(Y_k) \le (1 + \frac{\epsilon}{n})(\sum_{Y_k} d(Y_k) - d(u, Y(u)) - d(v, Y(v)) + d(u, Y(v)) + d(v, Y(u)) - 2d(u,v))$$

$$\le (1 + \frac{\epsilon}{n})(\sum_{Y_k} d(Y_k) - d(u, Y(u)) - d(v, Y(v)) + d(u, Y(v)) + d(v, Y(u))).$$

After moving terms and some simplification, we get the following inequality:

$$
\begin{aligned}
d(u, Y(u)) &+ d(v, Y(v)) \\
&\le d(u, Y(v)) + d(v, Y(u)) + \frac{\epsilon/n}{1 + \epsilon/n}\sum_{Y_k \in \mathcal{Y}} d(Y_k) \\
&\le d(u, Y(v)) + d(v, Y(u)) + \frac{\epsilon}{n}\sum_{Y_k \in \mathcal{Y}} d(Y_k).
\end{aligned}
\tag{4}
$$

Then we sum up (4), $d(u, Y(u)) + d(v, Y(v)) \le d(u, Y(v)) + d(v, Y(u)) + \frac{\epsilon}{n}\sum_{Y_k \in \mathcal{Y}} d(Y_k)$, over every pair of points in $red(V)$ (even if they are in the same partition).

$$r_t \sum_{Y_i} d_R(Y_i) \le \left(\sum_{Y_i} r_i d_R(Y_i)\right) + \left(\sum_{u \in red(V)} \sum_{Y_i \ne Y(u)} r_i d(u, Y_i)\right) + r_t^2 \frac{\epsilon}{n}\sum_{Y_i} d(Y_i).$$

Divide both sides by $r_t$ and use the fact that $r_i \le r$ for all $Y_i$:

$$\sum_{Y_i} d_R(Y_i) \le \left(\sum_{Y_i} \frac{r}{r_t} d_R(Y_i)\right) + \left(\sum_{u \in red(V)} \sum_{Y_i \ne Y(u)} \frac{r}{r_t} d(u, Y_i)\right) + \frac{r_t \epsilon}{n}\sum_{Y_i} d(Y_i). \tag{5}$$

For pairs of points in $blue(V)$ we sum (4) to similarly obtain:

$$\sum_{Y_i} d_B(Y_i) \le \left(\sum_{Y_i} \frac{b}{b_t} d_B(Y_i)\right) + \left(\sum_{v \in blue(V)} \sum_{Y_i \ne Y(v)} \frac{b}{b_t} d(v, Y_i)\right) + \frac{b_t \epsilon}{n}\sum_{Y_i} d(Y_i). \tag{6}$$

Now we sum up (5) and (6). The LHS becomes:

$$\sum_{Y_i}(d_R(Y_i) + d_B(Y_i)) = \sum_{Y_i}\sum_{u\in Y_i} d(u, Y_i) = 2\sum_{Y_i} d(Y_i.)$$

For the RHS, the last term in (5) and (6) is $\frac{\epsilon(b_t+r_t)}{n}\sum_{Y_i} d(Y_i) = \epsilon \sum_{Y_i} d(Y_i)$.
The other terms give:

$$\frac{r}{r_t}\sum_{Y_i} d_R(Y_i) + \frac{r}{r_t}\sum_{u\in red(V)}\sum_{Y_i\neq Y(u)} d(u, Y_i) + \frac{b}{b_t}\sum_{Y_i} d_B(Y_i) + \frac{b}{b_t}\sum_{v\in blue(V)}\sum_{Y_i\neq Y(v)} d(v, Y_i)$$

$$\leq \max\{\frac{r}{r_t}, \frac{b}{b_t}\}\left\{\sum_{Y_i}(d_R(Y_i) + d_B(Y_i)) + \sum_{u\in V}\sum_{Y_i\neq Y(u)} d(u, Y_i)\right\}$$

$$= \max\{\frac{r}{r_t}, \frac{b}{b_t}\}\left\{\sum_{Y_i}\sum_{u\in Y_i} d(u, Y_i) + \sum_{Y_i}\sum_{Y_j\neq Y_i} d(Y_i, Y_j)\right\}$$

$$= 2\max\{\frac{r}{r_t}, \frac{b}{b_t}\}d(V)$$

$$\leq \frac{4(b+r)}{n}d(V).$$

The last inequality follows from Proposition 22. All together, this proves that

$$2\sum_{Y_k} d(Y_k) \leq \frac{4(b+r)}{n}d(V) + \epsilon \sum_{Y_k} d(Y_k).$$

Then, $\frac{\sum_{Y_k} d(Y_k)}{d(V)} \leq \frac{2(b+r)}{n}\cdot\frac{1}{1-\epsilon/2} \leq (1+\epsilon)\frac{2(b+r)}{n}$. The final step follows from the fact that $(1+\epsilon)(1-\epsilon/2) = 1 + \frac{\epsilon}{2}(1-\epsilon) \geq 1$. This proves the lemma. $\square$

## C.2 Proof for the generalized Lemma 15

Next, we prove Lemma 15 for the more generalized definition of fairness, which is $\alpha$-capped fairness.

**Proof of** [Lemma 15] The proof follows the same logic as in the two-color case: we first use the $(\epsilon/n)$-local optimality of the solution, and sum up the inequality over all pairs of points with the same color.

Let $Y : V \mapsto \mathcal{Y}$ denote a mapping from a point in $V$ to the fairlet it belongs to. Let $R_i(X)$ be the set of $R_i$ colored points in a set $X$. Let $d_{R_i}(X) = \sum_{u\in R_i(X)} d(u, X)$. Naturally, $\sum_i d_{R_i}(x) = 2d(X)$ for any set $X$ since the weight for every pair of points is repeated twice.

The **While** loop can end in two cases: 1) if $\mathcal{Y}$ is $(\epsilon/n)$-locally-optimal; 2) if $\sum_{Y_k\in\mathcal{Y}} d(Y_k) \leq \Delta$. Case 2 immediately implies the lemma, thus we focus on case 1.

By definition of the algorithm, we know that for any pair $u \in Y(u)$ and $v \in Y(v)$ where $u, v$ have the same color and $Y(u) \neq Y(v)$ the swap does not increase objective value by a large amount. (The same trivially holds if the pair are in the same cluster.) We get the following inequality as in the two color case:

$$d(u, Y(u)) + d(v, Y(v)) \leq d(u, Y(v)) + d(v, Y(u)) + \frac{\epsilon}{n}\sum_{Y_k\in\mathcal{Y}} d(Y_k). \tag{7}$$

For any color $R_i$, we sum it over every pair of points in $R_i(V)$ (even if they are in the same partition).

$$n_i\sum_{Y_k} d_{R_i}(Y_k) \leq \left(\sum_{Y_k} r_{ik}d_{R_i}(Y_k)\right) + \left(\sum_{u\in R_i(V)}\sum_{Y_k\neq Y(u)} r_{ik}d(u, Y_k)\right) + n_i^2\frac{\epsilon}{n}\sum_{Y_k} d(Y_k).$$

Divide both sides by $n_i$ and we get:

$$\sum_{Y_k} d_{R_i}(Y_k) \leq \left(\sum_{Y_k} \frac{r_{ik}}{n_i}d_{R_i}(Y_k)\right) + \left(\sum_{u\in R_i(V)}\sum_{Y_k\neq Y(u)} \frac{r_{ik}}{n_i}d(u, Y_k)\right) + \frac{n_i\epsilon}{n}\sum_{Y_k} d(Y_k). \tag{8}$$

Now we sum up this inequality over all colors $R_i$. The LHS becomes:

$$\sum_{Y_k} \sum_i d_{R_i}(Y_k) = \sum_{Y_k} \sum_{u \in Y_k} d(u, Y_k) = 2 \sum_{Y_k} d(Y_k).$$

For the RHS, the last term sums up to $\frac{\epsilon(\sum_i n_i)}{n} \sum_{Y_k} d(Y_k) = \epsilon \sum_{Y_k} d(Y_k)$. Using the fact that $\frac{r_{ik}}{n_i} \leq \max_{i,k} \frac{r_{ik}}{n_i}$, the other terms sum up to :

$$\sum_i \sum_{Y_k} \frac{r_{ik}}{n_i} d_{R_i}(Y_k) + \sum_i \sum_{u \in R_i(V)} \sum_{Y_k \neq Y(u)} \frac{r_{ik}}{n_i} d(u, Y_k)$$

$$\leq \max_{i,k} \frac{r_{ik}}{n_i} \left\{ \sum_{Y_k} \sum_i d_{R_i}(Y_i) + \sum_{u \in V} \sum_{Y_k \neq Y(u)} d(u, Y_k) \right\}$$

$$= \max_{i,k} \frac{r_{ik}}{n_i} \left\{ \sum_{Y_k} \sum_{u \in Y_k} d(u, Y_k) + \sum_{Y_k} \sum_{Y_j \neq Y_k} d(Y_j, Y_k) \right\}$$

$$= 2 \max_{i,k} \frac{r_{ik}}{n_i} \cdot d(V).$$

Therefore, putting LHS and RHS together, we get

$$2 \sum_{Y_k} d(Y_k) \leq 2 \max_{i,k} \frac{r_{ik}}{n_i} d(V) + \epsilon \sum_{Y_k} d(Y_k).$$

Then, $\frac{\sum_{Y_k} d(Y_k)}{d(V)} \leq \max_{i,k} \frac{r_{ik}}{n_i} \cdot \frac{1}{1-\epsilon/2} \leq (1+\epsilon) \cdot \max_{i,k} \frac{r_{ik}}{n_i}$. The final step follows from the fact that $(1+\epsilon)(1-\epsilon/2) = 1 + \frac{\epsilon}{2}(1-\epsilon) \geq 1$.

$\square$

In the two-color case, the ratio $\max_{i,k} \frac{r_{ik}}{n_i}$ becomes $\max\{\frac{r}{r_t}, \frac{b}{b_t}\}$, which can be further bounded by $\frac{2(b+r)}{n}$ (see Proposition 22). If there exists a caplet decomposition such that $\max_{i,k} \frac{r_{ik}}{n_i} = o(1)$, Lemma 15 implies we can build a fair hierarchical clustering tree with $o(1)$ loss in approximation ratio for value objective.

Assuming for all color class $R_i$, $n_i \to +\infty$ as $n \to +\infty$, here we give a possible caplet decomposition for $\alpha = \frac{1}{t}(t <= c)$ with size $O(t)$ for positive integer $t$, thus guaranteeing $\max_{i,k} \frac{r_{ik}}{n_i} = o(1)$ for any $i$.

**Lemma 24.** *For any set $P$ of size $p$ that satisfies fairness constraint with $\alpha = 1/t$, there exists a partition of $P$ into sets $(P_1, P_2, \ldots)$ where each $P_i$ satisfies the fairness constraint and $t \leq |P_i| < 2t$.*

*Proof.* Let $p = m \times t + r$ with $0 \leq r < t$, then the fairness constraints ensures that there are at most $m$ elements of each color. Consider partitioning obtained through the following process: consider an ordering of elements where points of the same color are in consecutive places, assign points to sets $P_1, P_2, \ldots, P_m$ in a round robin fashion. So each set $P_i$ gets at least $t$ elements and at most $t + r < 2t$ elements assigned to it. Since there are at most $m$ elements of each color, each set gets at most one point of any color and hence all sets satisfy the fairness constraint as $1 \leq \frac{1}{t} \cdot |P_i|$. $\square$

### C.3 Proof for the running time of $(\epsilon/n)$-locally-optimal fairlet decomposition algorithm

**Proof of** [Lemma 16] Notice that finding the maximum pairwise distance takes $O(n^2)$ time. Thus, we focus on analyzing the time spent on the **While** loop.

Let $t$ be the total number of swaps. We argue that $t = \tilde{O}(n/\epsilon)$. If $t = 0$ the conclusion trivially holds. Otherwise, consider the decomposition $\mathcal{Y}_{t-1}$ before the last swap. Since the **While** loop does not terminate here, $\sum_{Y_k \in \mathcal{Y}_{t-1}} d(Y_k) \geq \Delta = \frac{b+r}{n} d_{max}$. However, at the beginning, we have $\sum_{Y_k \in \mathcal{Y}} d(Y_k) \leq (b+r)n \cdot d_{max} = n^2 \Delta \leq n^2 \sum_{Y_k \in \mathcal{Y}_{t-1}} d(Y_k)$. Therefore, it takes at most $\log_{1+\epsilon/n}(n^2) = \tilde{O}(n/\epsilon)$ iterations to finish the **While** loop.

It remains to discuss the running time of each iteration. We argue that there is a way to finish each iteration in $O(n^2)$ time. Before the **While** loop, keep a record of $d(u, Y_i)$ for each point $u$ and each fairlet $Y_i$. This takes $O(n^2)$ time. If we know $d(u, Y_i)$ and the objective value from the last iteration, in the current iteration, it takes $O(1)$ time to calculate the new objective value after each swap $(u, v)$, and there are at most $n^2$ such calculations, before the algorithm either finds a pair to swap, or determines that no such pair is left. After the swap, the update for all the $d(u, Y_i)$ data takes $O(n)$ time. In total, every iteration takes $O(n^2)$ time.

Therefore, Algorithm 1 takes $\tilde{O}(n^3/\epsilon)$ time. $\qquad\square$

## D  Hardness of optimal fairlet decomposition

Before proving Theorem 7, we state that the PARTITION INTO TRIANGLES (PIT) problem is known to belong to the NP-complete class [23], defined as follows. In the definition, we call a clique $k$-clique if it has $k$ nodes. A triangle is a 3-clique.

**Definition 25.** PARTITION INTO TRIANGLES
(PIT). *Given graph $G = (V, E)$, where $V = 3n$, determine if $V$ can be partitioned into 3-element sets $S_1, S_2, \ldots, S_n$, such that each $S_i$ forms a triangle in $G$.*

The NP-hardness of PIT problem gives us a more general statement.

**Definition 26.** PARTITION INTO $k$-CLIQUES
(PIKC). *For a fixed number $k$ treated as constant, given graph $G = (V, E)$, where $V = kn$, determine if $V$ can be partitioned into $k$-element sets $S_1, S_2, \ldots, S_n$, such that each $S_i$ forms a $k$-clique in $G$.*

**Lemma 27.** *For a fixed constant $k \geq 3$, the PIKC problem is NP-hard.*

*Proof.* We reduce the PIKC problem from the PIT problem. For any graph $G = (V, E)$ given to the PIT problem where $|V| = 3n$, construct another graph $G' = (V', E')$. Let $V' = V \cup C_1 \cup C_2 \cup \cdots \cup C_n$, where all the $C_i$'s are $(k-3)$-cliques, and there is no edge between any two cliques $C_i$ and $C_j$ where $i \neq j$. For any $C_i$, let all points in $C_i$ to be connected to all nodes in $V$.

Now let $G'$ be the input to PIKC problem. We prove that $G$ can be partitioned into triangles if and only if $G'$ can be partitioned into $k$-cliques. If $V$ has a triangle partition $V = \{S_1, \ldots, S_n\}$, then $V' = \{S_1 \cup C_1, \ldots, S_n \cup C_n\}$ is a $k$-clique partition. On the other hand, if $V'$ has a $k$-clique partition $V' = \{S_1', \ldots, S_n'\}$ then $C_1, \ldots, C_n$ must each belong to different $k$-cliques since they are not connected to each other. Without loss of generality we assume $C_i \subseteq S_i$, then $V = \{S_1' \setminus C_1, \ldots, S_n' \setminus C_n\}$ is a triangle partition. $\qquad\square$

We are ready to prove the theorem.

**Proof of** [Theorem 7] We prove Theorem 7 by proving that for given $z \geq 4$, if there exists a $c$-approximation polynomial algorithm $\mathcal{A}$ for (3), it can be used to solve the PIKC problem where $k = z - 1$ for any instance as well. This holds for any finite $c$.

Given any graph $G = (V, E)$ that is input to the PIKC problem, where $|V| = kn = (z-1)n$, let a set $V'$ with distances be constructed in the following way:

1. $V' = V \cup \{C_1, \ldots, C_n\}$, where each $C_i$ is a singleton.
2. Color the points in $V$ red, and color all the $C_i$'s blue.
3. For a $e = (u, v)$, let $d(u, v) = 0$, if it satisfies one of the three conditions: 1) $e \in E$. 2) $u, v \in C_i$ for some $C_i$. 3) one of $u, v$ is in $V$, while the other belong to some $C_i$.
4. All other edges have distance 1.

Obviously the blue points make up a $1/z$ fraction of the input so each fairlet should have exactly 1 blue point and $z - 1$ red points.

We claim that $G$ has a $k$-clique partition if and only if algorithm $\mathcal{A}$ gives a solution of 0 for (3). The same argument as in the proof of Lemma 27 will show that $G$ has a $k$-clique partition if and only if the optimal solution to (3) is 0. This is equal to algorithm $\mathcal{A}$ giving a solution of 0 since otherwise the approximate is not bounded. $\qquad\square$

# E Optimizing cost with fairness

In this section, we present our fair hierarchical clustering algorithm that approximates Dasgupta's cost function and satisfies Theorem 17. Most of the proofs can be found in Section E.1. We consider the problem of equal representation, where vertices are red or blue and $\alpha = 1/2$. From now on, whenever we use the word "fair", we are referring to this fairness constraint. Our algorithm also uses parameters $t$ and $\ell$ such that $n \geq t\ell$ and $t > \ell + 108t^2/\ell^2$ for $n = |V|$, and leverages a $\beta$-approximation for cost and $\gamma_t$-approximation for minimum weighted bisection. We will assume these are fixed and use them throughout the section.

We will ultimately show that we can find a fair solution that is a sublinear approximation for the unfair optimum $T^*_{\text{unfair}}$, which is a lower bound of the fair optimum. Our main result is Theorem 17, which is stated in the body of the paper.

The current best approximations described in Theorem 17 are $\gamma_t = O(\log^{3/2} n)$ by [22] and $\beta = \sqrt{\log n}$ by both [20] and [12]. If we set $t = \sqrt{n}(\log^{3/4} n)$ and $\ell = n^{1/3}\sqrt{\log n}$, then we get Corollary 28.

**Corollary 28.** *Consider the equal representation problem with two colors. There is an $O\left(n^{5/6} \log^{5/4} n\right)$-approximate fair clustering under the cost objective.*

The algorithm will be centered around a single clustering, which we call $\mathcal{C}$, that is extracted from an unfair hierarchy. We will then adapt this to become a similar, fair clustering $\mathcal{C}'$. To formalize what $\mathcal{C}'$ must satisfy to be sufficiently "similar" to $\mathcal{C}$, we introduce the notion of a $\mathcal{C}$-good clustering. Note that this is not an intuitive set of properties, it is simply what $\mathcal{C}'$ must satisfy in order

**Definition 29** (Good clustering). *Fix a clustering $\mathcal{C}$ whose cluster sizes are at most $t$. A fair clustering $\mathcal{C}'$ is $\mathcal{C}$-good if it satisfies the following two properties:*

1. *For any cluster $C \in \mathcal{C}$, there is a cluster $C' \in \mathcal{C}'$ such that all but (at most) an $O(\ell\gamma_t/t + t\gamma_t/\ell^2)$-fraction of the weight of edges in $C$ is also in $C'$.*
2. *Any $C' \in \mathcal{C}'$ is not too much bigger, so $|C'| \leq 6t\ell$.*

The hierarchy will consist of a $\mathcal{C}$-good (for a specifically chosen $\mathcal{C}$) clustering $\mathcal{C}'$ as its only nontrivial layer.

**Lemma 30.** *Let $T$ be a $\beta$-approximation for cost and $\mathcal{C}$ be a maximal clustering in $T$ under the condition that all cluster sizes are at most $t$. Then, a fair two-tiered hierarchy $T'$ whose first level consists of a $\mathcal{C}$-good clustering achieves an $O\left(\frac{n}{t} + t\ell + \frac{n\ell\gamma_t}{t} + \frac{nt\gamma_t}{\ell^2}\right)\beta$-approximation for cost.*

*Proof.* Since $T$ is a $\beta$-approximation, we know that:

$$\text{cost}(T) \leq \beta\text{cost}(T^*_{\text{unfair}})$$

We will then utilize a scheme to account for the cost contributed by each edge relative to their cost in $T$ in the hopes of extending it to $T^*_{\text{unfair}}$. There are three different types of edges:

1. An edge $e$ that is merged into a cluster of size $t$ or greater in $T$, thus contributing $t \cdot s(e)$ to the cost. At worst, this edge is merged in the top cluster in $T'$ to contribute $n \cdot s(e)$. Thus, the factor increase in the cost contributed by $e$ is $O(n/t)$. Then since the total contribution of all such edges in $T$ is at most $\text{cost}(T)$, the total contribution of all such edges in $T'$ is at most $O(n/t) \cdot \text{cost}(T)$.
2. An edge $e$ that started in some cluster $C \in \mathcal{C}$ that does not remain in the corresponding cluster $C'$. We are given that the total weight removed from any such $C$ is an $O(\ell\gamma_t/t + t\gamma_t/\ell^2)$-fraction of the weight contained in $C$. If we sum across the weight in all clusters in $\mathcal{C}$, that is at most $\text{cost}(T)$. So the total amount of weight moved is at most $O(\ell\gamma_t/t + t\gamma_t/\ell^2) \cdot \text{cost}(T)$. These edges contributed at least $2s(e)$ in $T$ as the smallest possible cluster size is two. In $T'$, these may have been merged at the top of the cluster, for a maximum cost contribution of $n \cdot s(e)$. Therefore, the total cost across all such edges is increased by at most a factor of $n/2$, which gives a total cost of at most $O(n\ell\gamma_t/t + nt\gamma_t/\ell^2) \cdot \text{cost}(T)$.
3. An edge $e$ that starts in some cluster $C \in \mathcal{C}$ and remains in the corresponding $C' \in \mathcal{C}'$. Similarly, this must have contributed at least $2s(e)$ in $T$, but now we know that this edge is

merged within $C'$ in $T'$, and that the size of $C'$ is $|C'| \leq 6t\ell$. Thus its contribution increases at most by a factor of $3t\ell$. By the same reasoning from the first edge type we discussed, all these edges total contribute at most a factor of $O(t\ell) \cdot \text{cost}(T)$.

We can then put a conservative bound by putting this all together.

$$\text{cost}(T') \leq O\left(\frac{n}{t} + t\ell + \frac{n\ell\gamma_t}{t} + \frac{nt\gamma_t}{\ell^2}\right)\text{cost}(T).$$

Finally, we know $T$ is a $\beta$-approximation for $T^*_{\text{unfair}}$.

$$\text{cost}(T') \leq O\left(\frac{n}{t} + t\ell + \frac{n\ell\gamma_t}{t} + \frac{nt\gamma_t}{\ell^2}\right)$$
$$\cdot \beta \cdot \text{cost}(T^*_{\text{unfair}}).\square$$

With this proof, the only thing left to do is find a $\mathcal{C}$-good clustering $\mathcal{C}'$ (Definition 29). Specifically, using the clustering $\mathcal{C}$ mentioned in Lemma 30, we would like to find a $\mathcal{C}$-good clustering $\mathcal{C}'$ using the following.

**Lemma 31.** *There is an algorithm that, given a clustering $\mathcal{C}$ with maximum cluster size $t$, creates a $\mathcal{C}$-good clustering.*

The proof is deferred to the Section E.1. With these two Lemmas, we can prove Theorem 17.

*Proof.* Consider our graph $G$. We first obtain a $\beta$-approximation for unfair cost, which yields a hierarchy tree $T$. Let $\mathcal{C}$ be the maximal clustering in $T$ under the constraint that the cluster sizes must not exceed $t$. We then apply the algorithm from Lemma 31 to get a $\mathcal{C}$-good clustering $\mathcal{C}'$. Construct $T'$ such that it has one layer that is $\mathcal{C}'$. Then we can apply the results from Lemma 30 to get the desired approximation. $\square$

From here, we will only provide a high-level description of the algorithm for Lemma 31. For precise details and proofs, see Section E.1. To start, we need to propose some terminology.

**Definition 32** (Red-blue matching). *A **red-blue matching** on a graph $G$ is a matching $M$ such that $M(u) = v$ implies $u$ and $v$ are different colors.*

Red-blue matchings are interesting because they help us ensure fairness. For instance, suppose $M$ is a red-blue matching that is also perfect (i.e., touches all nodes). If the lowest level of a hierarchy consists of a clustering such that $v$ and $M(v)$ are in the same cluster for all $v$, then that level of the hierarchy is fair since there is a bijection between red and blue vertices within each cluster. When these clusters are merged up in the hierarchy, fairness is preserved.

Our algorithm will modify an unfair clustering to be fair by combining clusters and moving a small number of vertices. To do this, we will use the following notion.

**Definition 33** (Red-blue clustering graph). *Given a graph $G$ and a clustering $\mathcal{C} = \{C_1, \ldots, C_k\}$, we can construct a **red-blue clustering graph** $H_M = (V_M, E_M)$ that is associated with some red-blue matching $M$. Then $H_M$ is a graph where $V_M = \mathcal{C}$ and $(C_i, C_j) \in E_M$ if and only if there is a $v_i \in C_i$ and $M(v_i) = v_j \in C_j$.*

Basically, we create a graph of clusters, and there is an edge between two clusters if and only if there is at least one vertex in one cluster that is matched to some vertex in the other cluster. We now show that the red-blue clustering graph can be used to construct a fair clustering based on an unfair clustering.

**Proposition 34.** *Let $H_M$ be a red-blue clustering graph on a clustering $\mathcal{C}$ with a perfect red-blue matching $M$. Let $\mathcal{C}'$ be constructed by merging all the clusters in each component of $H_M$. Then $\mathcal{C}'$ is fair.*

*Proof.* Consider some $C \in \mathcal{C}'$. By construction, this must correspond to a connected component in $H_M$. By definition of $H_M$, for any vertex $v \in C$, $M(v) \in C$. That means $M$, restricted to $C$, defines a bijection between the red and blue nodes in $C$. Therefore, $C$ has an equal number of red and blue vertices and hence is fair. $\square$

We will start by extracting a clustering $\mathcal{C}$ from an unfair hierarchy $T$ that approximates cost. Then, we will construct a red-blue clustering graph $H_M$ with a perfect red-blue matching $M$. Then we can use the components of $H_M$ to define our first version of the clustering $\mathcal{C}'$. However, this requires a non-trivial way of moving vertices between clusters in $\mathcal{C}$.

We now give an overview of our algorithm in Steps (A)–(G). For a full description, see our pseudocode in Section H.

(A) **Get an unfair approximation** $T$. We start by running a $\beta$-approximation for cost in the unfair setting. This gives us a tree $T$ such that $\mathrm{cost}(T) \leq \beta \cdot \mathrm{cost}(T^*_{\mathrm{unfair}})$.

(B) **Extract a $t$-maximal clustering**. Given $T$, we find the maximal clustering $\mathcal{C}$ such that (i) every cluster in the clustering is of size at most $t$, and (ii) any cluster above these clusters in $T$ is of size more than $t$.

(C) **Combine clusters to be size $t$ to $3t$.** We will now slowly change $\mathcal{C}$ into $\mathcal{C}'$ during a number of steps. In the first step, we simply define $\mathcal{C}_0$ by merging small clusters $|C| \leq t$ until the merged size is between $t$ and $3t$. Thus clusters in $\mathcal{C}$ are contained within clusters in $\mathcal{C}_0$, and all clusters are between size $t$ and $3t$.

(D) **Find cluster excesses.** Next, we strive to make our clustering more fair. We do this by trying to find an underlying matching between red and blue vertices that agrees with $\mathcal{C}_0$ (matches are in the same cluster). If the matching were perfect, then the clusters in $\mathcal{C}_0$ would have equal red and blue representation. However, this is not guaranteed initially. We start by conceptually matching as many red and blue vertices within clusters as we can. Note we do not actually create this matching; we just want to reserve the space for this matching to ensure fairness, but really some of these vertices may be moved later on. Then the remaining unmatched vertices in each cluster is either entirely red or entirely blue. We call this amount the *excess* and the color the *excess color*. We label each cluster with both of these.

(E) **Construct red-blue clustering graph**. Next, we would like to construct $H_M = (V_M, E_M)$, our red-blue clustering graph on $\mathcal{C}_0$. Let $V_M = \mathcal{C}_0$. In addition, for the within-cluster matchings mentioned in Step (D), let those matches be contained in $M$. With this start, we will do a matching process to simultaneously construct $E_M$ and the rest of $M$. Note the unmatched vertices are specifically the excess vertices in each cluster. We will match these with an iterative process given our parameter $\ell$:

1. Select a vertex $C_i \in V_M$ with excess at least $\ell$ to start a new connected component in $H_M$. Without loss of generality, say its excess color is red.
2. Find a vertex $C_j \in V_M$ whose excess color is blue and whose excess is at least $\ell$. Add $(C_i, C_j)$ to $E_M$.
3. Say without loss of generality that the excess of $C_i$ is less than that of $C_j$. Then match all the excess in $C_i$ to vertices in the excess of $C_j$. Now $C_j$ has a smaller excess.
4. If $C_j$ has an excess less than $\ell$ or $C_j$ is the $\ell$th cluster in this component, end this component. Start over at (1) with a new cluster.
5. Otherwise, use $C_j$ as our reference and continue constructing this component at (2).
6. Complete when there are no more clusters with over $\ell$ excess that are not in a component (or all remaining such clusters have the same excess color).

We would like to construct $\mathcal{C}'$ by merging all clusters in each component. This would be fair if $M$ were a perfect matching, however this is not true yet. In the next step, we handle this.

(F) **Fix unmatched vertices.** We now want to match excess vertices that are unmatched. We do this by bringing vertices from other clusters into the clusters that have unmatched excess, starting with all small unmatched excess. Note that some clusters were never used in Step (E) because they had small excess to start. This means they had many internal red-blue matches. Remove $t^2/\ell^2$ of these and put them into clusters in need. For other vertices, we will later describe a process where $t/\ell$ of the clusters can contribute $108t^2/\ell^2$ vertices to account for unmatched excess. Thus clusters lose at most $108t^2/\ell^2$ vertices, and we account for all unmatched vertices. Call the new clustering $\mathcal{C}_1$. Now $M$ is perfect and $H_M$ is unchanged.

(G) **Define $\mathcal{C}'$.** Finally, we create the clustering $\mathcal{C}'$ by merging the clusters in each component of $H_M$. Note that Proposition 34 assures $\mathcal{C}'$ is fair. In addition, we will show that cluster sizes in $\mathcal{C}_1$

are at most $6t$, so $\mathcal{C}'$ has the desired upper bound of $6t\ell$ on cluster size. Finally, we removed at most $\ell + t^2/\ell^2$ vertices from each cluster. This is the desired $\mathcal{C}$-good clustering.

Further details and the proofs that the above sequence of steps achieve the desired approximation can be found in the next section. While the approximation factor obtained is not as strong as the ones for revenue or value objectives with fairness, we believe cost is a much harder objective with fairness constraints.

### E.1   Proof of Theorem 17

This algorithm contains a number of components. We will discuss the claims made by the description step by step. In Step (A), we simply utilize any $\beta$-approximation for the unfair approximation. Step (B) is also quite simple. At this point, all that is left is to show how to find $\mathcal{C}'$, ie, prove Lemma 31 (introduced in Section 6). This occurs in the steps following Step (B). In Step (C), we apply our first changes to the starting clustering from $T$. We now prove that the cluster sizes can be enforced to be between $t$ and $3t$.

**Lemma 35.** *Given a clustering $\mathcal{C}$, we can construct a clustering $\mathcal{C}_0$, where each $C \in \mathcal{C}_0$ is a union of clusters in $\mathcal{C}$ and $t \le |C| < 3t$.*

*Proof.* We iterate over all clusters in $\mathcal{C}$ whose size are less than $t$ and continually merge them until we create a cluster of size $\ge t$. Note that since the last two clusters we merged were of size $< t$, this cluster is of size $t \le |C| < 2t$. We then stop this cluster and continue merging the rest of the clusters. At the end, if we are left with a single cluster of size $< t$, we simply merge this with any other cluster, which will then be of size $t \le |C| < 3t$. $\qquad\square$

Step (D) describes a rather simple process. All we have to do in each cluster is count the amount of each color in each cluster, find which is more, and also compute the difference. No claims are made here.

Step (E) defines a more careful process. We describe this process and its results here.

**Lemma 36.** *There is an algorithm that, given a clustering $\mathcal{C}_0$ with $t \le |C| \le 3t$ for $C \in \mathcal{C}_0$, can construct a red-blue clustering graph $H_M = (V_M, E_M)$ on $\mathcal{C}_0$ with underlying matching $M$ such that:*

1. *$H_M$ is a forest, and its max component size is $\ell$.*

2. *For every $(C_i, C_j) \in E_M$, there are at least $\ell$ matches between $C_i$ and $C_j$ in $M$. In other words, $|M(C_i) \cap C_j| \ge \ell$.*

3. *For most $C_i \in V_M$, at most $\ell$ vertices in $C_i$ are unmatched in $M$. The only exceptions to this rule are (1) exactly one cluster in every $\ell$-sized component in $H_M$, and (2) at most $n/2$ additional clusters.*

*Proof.* We use precisely the process from Step 5. Let $V_M = \mathcal{C}_0$. $H_M$ will look like a bipartite graph with some entirely isolated nodes. We then try to construct components of $H_M$ one-by-one such that (1) the max component size is $\ell$, and (2) edges represent at least $\ell$ matches in $M$.

Let us show it satisfies the three conditions of the lemma. For condition 1, note that we will always halt component construction once it reaches size $\ell$. Thus no component can exceed size $\ell$. In addition, for every edge added to the graph, at least one of its endpoints now has small excess and will not be considered later in the program. Thus no cycles can be created, so it is a forest.

For condition 2, consider the construction of any edge $(C_i, C_j) \in E_M$. At this point, we only consider $C_i$ and $C_j$ to be clusters with different-color excess of at least $\ell$ each. In the next part of the algorithm, we match as much excess as we can between the two clusters. Therefore, there must be at least $\ell$ underlying matches.

Finally, condition 3 will be achieved by the completion condition. By the completion condition, there are no isolated vertices (besides possibly those leftover of the same excess color) that have over $\ell$ excess. Whenever we add a cluster to a component, either that cluster matches all of its excess, or the cluster it becomes adjacent to matches all of its excess. Therefore at any time, any component

has at most one cluster with any excess at all. If the component is smaller than $\ell$ (and is not the final component), then that can only happen when in the final addition, both clusters end up with less than $\ell$ excess. Therefore, no cluster in this component can have less than $\ell$ excess. For an $\ell$-sized component, by the rule mentioned before, only one cluster can remain with $\ell$ excess. When the algorithm completes, we are left with a number of large-excess clusters with the same excess color, say red without loss of generality. Assume for contradiction there are more than $n/2$ such clusters, and so there is at least $n\ell/2$. Since we started with half red and half blue vertices, the remaining excess in the rest of the clusters must match up with the large red excess. Thus the remaining at most $n/2$ clusters must have at least $n\ell/2$ blue excess, but this is only achievable if they have large excess left. This is a contradiction. Thus we satisfy condition 3. □

This concludes Step (E). In Step (F), we will transform the underlying clustering $\mathcal{C}_0$ such that we can achieve a perfect matching $M$. This will require removing a small number of vertices from some clusters in $\mathcal{C}_0$ and putting them in clusters that have unmatched vertices. This process will at most double cluster size.

**Lemma 37.** *There is an algorithm that, given a clustering $\mathcal{C}_0$ with $t \leq |C| \leq 3t$ for $C \in \mathcal{C}_0$, finds a clustering $\mathcal{C}_1$ and an underlying matching $M'$ such that:*

1. *There is a bijection between $\mathcal{C}_0$ and $\mathcal{C}_1$.*

2. *For any cluster $C_0 \in \mathcal{C}_0$ and its corresponding $C_1 \in \mathcal{C}_1$, $|C_0| - |C_1| \leq \ell + 108t^2/\ell^2$. This means that at most $\ell$ vertices are removed from $C_0$ in the construction of $C_1$.*

3. *For all $C_1 \in \mathcal{C}_1$, $t - \ell - 108t^2/\ell^2 \leq |C_1| \leq 6t$.*

4. *$M'$ is a perfect red-blue matching.*

5. *$H_M$ is a red-blue clustering graph of $\mathcal{C}_1$ with matching $M'$, perhaps with additional edges.*

*Proof.* Use Lemma 36 to find the red-blue clustering graph $H_M$ and its corresponding graph $M$. Then we know that only one cluster in every $\ell$-sized component plus one other cluster can have a larger than $\ell$ excess. Since cluster sizes are at least $t$, $|V_M| \geq n/t$. This means that at most $n/(t\ell) + 1 = (n + t\ell)/(t\ell) \leq 2n/(t\ell)$ clusters need more than $\ell$ vertices. Since the excess is upper bounded by cluster size which is upper bounded by $3t$, this is at most $6n/\ell$ vertices in large excess that need matches.

We will start by removing all small excess vertices from clusters. This removes at most $\ell$ from any cluster. These vertices will then be placed in clusters with large excess of the right color. If we run out of large excess of the right color that needs matches, since the total amount of red and blue vertices is balanced, that means we can instead transfer the unmatched small excess red vertices to clusters with a small amount of unmatched blue vertices. In either case, this accounts for all the small unmatched excess. Now all we need to account for is at most $6n/\ell$ unmatched vertices in large excess clusters. At this point, note that the large excess should be balanced between red and blue. From now on, we will remove matches from within and between clusters to contribute to this excess. Since this always contributes the same amount of red and blue vertices by breaking matches, we do not have to worry about the balance of colors. We will describe how to distribute these contributions across a large number of clusters.

Consider vertices that correspond to clusters that (ignoring the matching $M$) started out with at most $\ell$ excess. So the non-excess portion, which is at least size $t - \ell$, is entirely matched with itself. We will simply remove $t^2/\ell^2$ of these matches to contribute.

Otherwise, we will consider vertices that started out with large excess. We must devise a clever way to break matches without breaking too many incident upon a single cluster. For every tree in $H_M$ (since $H_M$ is a forest by Lemma 36), start at the root, and do a breadth-first search over all internal vertices. At any vertex we visit, break $\ell$ matches between it and its child (recall by by Lemma 36 that each edge in $H_M$ represents at least $\ell$ inter-cluster matches). Thus, each break contributes $2\ell$ vertices. We do this for every internal vertex. Since an edge represents at least $\ell$ matches and the max cluster size is at most $3t$, any vertex can have at most $3t/\ell$ children. Thus the fraction of vertices in $H_M$ that correspond to a contribution of $2\ell$ vertices is at least $\ell/(3t)$.

Clearly, the worst case is when all vertices in $H_M$ have large excess, as this means that fewer clusters are ensured to be able to contribute. By Lemma 36, at least $n/2$ of these are a part of completed connected components (ie, of size $\ell$ or with each cluster having small remaining excess). So consider this case. Since $|V_M| \geq n/(3t)$, then this process yields $n\ell^2/(18t^2)$ vertices. To achieve $6n/\ell$ vertices, we must then run $108t^2/\ell^3$ iterations. If an edge no longer represents $\ell$ matches because of an earlier iteration, consider it a non-edge for the rest of the process. The only thing left to consider is if a cluster $C$ becomes isolated in $H_M$ during the process. We know $C$ began with at least $t$ vertices, and at most $\ell$ were removed by removing small excess. So as long as $t > \ell + 108t^2/\ell^2$, we can remove the rest of the $108t^2/\ell^2$ vertices from the non-excess in $C$ (the rest must be non-excess) in the same way as vertices that were isolated in $H_M$ to start. Thus, we can account for the entire set of unmatched vertices without removing more than $108t^2/\ell^2$ vertices from any given cluster.

Now we consider the conditions. Condition 1 is obviously satisfied because we are just modifying clusters in $\mathcal{C}_0$, not removing them. The second condition is true because of our careful accounting scheme where we only remove $\ell + 108t^2/\ell^2$ vertices per cluster. The same is true for the lower bound in condition 3. When we add them to new clusters, since we only add a vertex to match an unmatched vertex, we at most double cluster size. So the max cluster size is $6t$.

For the fourth condition, note that we explicitly executed this process until all unmatched vertices became matched, and any endpoint in a match we broke was used to create a new match. Thus the new matching, which we call $M'$, is perfect. It is still red-blue. Finally, note we did not create any matches between clusters. Therefore, no match in $M'$ can violate $H_M$. Thus condition 5 is met. $\square$

Finally, we construct our final clustering in Step (G). However, to satisfy the qualities of Lemma 30, we must first argue about the weight loss from each cluster.

**Lemma 38.** *Consider any clustering $\mathcal{C}$ with cluster sizes between $t$ and $6t$. Say each cluster has a specified $r$ number of red vertices to remove and $b$ number of blue vertices to remove such that $r + b \leq x$ for some $x$, and $r$ (resp. $b$) is nonzero only if the number of red (resp. blue) vertices in the cluster is $O(n)$. Then we can remove the desired number of each color while removing at most an $O((x/t)\gamma_t)$ of the weight originally contained within the cluster.*

*Proof.* Consider some cluster $C$ with parameters $r$ and $b$. We will focus first on removing red vertices. Let $C_r$ be the red vertex set in $C$. We create a graph $K$ corresponding to this cluster as follows. Let $b_0$ be a vertex representing all blue vertices from $C$, $b_0'$ be the "complement" vertex to $b_0$, and $R$ be a set of vertices $r_i$ corresponding to all red vertices in $C$. We also add a set of $2r - |C_r| + 2X$ dummy vertices (where $X$ is just some large value that makes it so $2r - |C_r| + X > 0$). $2r - |C_r| + X$ of the dummy vertices will be connected to $b_0$ with infinite edge weight (denote these $\delta_i$), the other $X$ will be connected to $b_0'$ with infinite edge weight (denote these $\delta_i'$). This will ensure that $b_0$ and $b_0'$ are in the same partitions as their corresponding dummies. Let $s_G$ and $s_K$ be the similarity function in the original graph and new graph respectively.

$$s_K(b_0, \delta_i) = \infty$$
$$s_K(b_0', \delta_i') = \infty$$

The blue vertex $b_0$ is also connected to all $r_i$ with the following weight (where $C_b$ is the set of blue vertices in $C$):

$$s_K(b_0, r_i) = \sum_{b_j \in C_b} s_G(r_i, b_j) + \frac{1}{2} \sum_{r_j \in R \setminus \{r_i\}} s_G(r_i, r_j)$$

This edge represents the cumulative edge weight between $r_i$ and all blue vertices. The additional summation term, which contains the edge weights between $r_i$ and all other red vertices, is necessary to ensure our bisection cut will also contain the edge weights between two of the removed red vertices.

Next, the edge weights between red vertices must contain the other portion of the corresponding edge weight in the original graph.

$$s_K(r_i, r_j) = \frac{1}{2} s_G(r_i, r_j)$$

Now, we note that there are a total of $2 - |C_r| + 2X + |C_r| = 2r + 2X$ vertices. So a bisection will partition the graph into vertex sets of size $r + X$. Obviously, in any approximation, $b_0$ must be grouped with all $\delta_i$ and $b_0'$ must be grouped with all $\delta_i'$. This means the $b_0$ partition must contain $|C_r| - r$ of the $R$ vertices, and the $b_0'$ partition must contain the other $r$. These $r$ vertices in the latter partition are the ones we select to move.

Consider any set $S$ of $r$ red vertices in $K$. Then it is a valid bisection. We now show that the edge weight in the cut for this bisection is exactly the edge weight lost by removing $S$ from $K$. We can do this algebraically. We start by breaking down the weight of the cut into the weight between the red vertices in $S$ and $b_0$, and also the red vertices in $S$ and the red vertices not in $S$.

$$
\begin{aligned}
s_K&(S, V(K) \setminus S) \\
&= \sum_{r_i \in S} s_K(b_0, r_i) + \sum_{r_i \in S, r_j \in R \setminus S} s_K(r_i, r_j) \\
&= \sum_{r_i \in S} \left( \sum_{b_j \in B} s_G(r_i, b_j) + \frac{1}{2} \sum_{r_j \in R \setminus \{r_j\}} s_G(r_i, r_j) \right) \\
&\quad + \sum_{r_i \in S, r_j \in R \setminus S} \frac{1}{2} s_G(r_i, r_j) \\
&= \sum_{r_i \in S} \left( \sum_{b_j \in B} s_G(r_i, b_j) + \frac{1}{2} \sum_{r_j \in R \setminus \{r_j\}} s_G(r_i, r_j) \right. \\
&\quad \left. + \frac{1}{2} \sum_{r_j \in R \setminus S} s_G(r_i, r_j) \right)
\end{aligned}
$$

Notice that the two last summations have an overlap. They both contribute half the edge weight between $r_i$ and vertices in $R \setminus S$. Thus, these edges contribute their entire edge weight. All remaining vertices in $S \setminus \{r_i\}$ only contribute half their edge weight. We can then distribute the summation.

$$
\begin{aligned}
s_K&(S, V(K) \setminus S) \\
&= \sum_{r_i \in S} \left( \sum_{b_j \in B} s_G(r_i, b_j) + \frac{1}{2} \sum_{r_j \in S \setminus \{r_j\}} s_G(r_i, r_j) \right. \\
&\quad \left. + \sum_{r_j \in R \setminus S} s_G(r_i, r_j) \right) \\
&= \sum_{r_i \in S, b_j \in B} s_G(r_i, b_j) + \frac{1}{2} \sum_{r_i \in S, r_j \in S \setminus \{r_j\}} s_G(r_i, r_j) \\
&\quad + \sum_{r_i \in S, r_j \in R \setminus S} s_G(r_i, r_j)
\end{aligned}
$$

In the middle summation, note that every edge $e = (u, v)$ is counted twice when $r_i = u$ and $r_j = v$, and when $r_i = v$ and $r_j = u$. We can then rewrite this as:

$$s_K(S, V(K) \setminus S) = \sum_{r_i \in S, b_j \in B} s_G(r_i, b_j)$$
$$+ \sum_{r_i, r_j \in S} s_G(r_i, r_j)$$
$$+ \sum_{r_i \in S, r_j \in R \setminus S} s_G(r_i, r_j)$$

When we remove $S$, we remove the connections between $S$ and blue vertices, the connections within $S$, and the connections between $S$ and red vertices not in $S$. This is precisely what this accounts for. Therefore, any bisection on $K$ directly corresponds to removing a vertex set $S$ of $r$ red vertices from $C$. If we have a $\gamma_t$-approximation for minimum weighted bisection, then, this yields a $\gamma_t$-approximation for the smallest loss we can achieve from removing $r$ red vertices.

Now we must compare the optimal way to remove $r$ vertices to the total weight in a cluster. Let $\rho = |C_r|$ be the number of red vertices in a cluster. Then the total number of possible cuts to isolate $r$ red vertices is $\binom{\rho}{r}$. Let $\mathcal{S}$ be the set of all possible cuts to isolate $r$ red vertices. Then if we sum over the weight of all possible cuts (where weight here is the weight between the $r$ removed vertices and all vertices, including each other), that will sum over each red-red edge and blue-red edge multiple times. A red-red edge is counted if either of its endpoints is in $S \in \mathcal{S}$, and this happens $2\binom{\rho}{r-1} - \binom{R-1}{r-2} \le 2\binom{\rho}{r-1}$ of the time. A blue-red edge is counted if its red endpoint is in $S$, which happens $\binom{\rho}{r-1} \le 2\binom{\rho}{r-1}$. And of course, since no blue-blue edge is covered, each is covered under $2\binom{\rho}{r-1}$ times. Therefore, if we sum over all these cuts, we get at most $2\binom{\rho}{r-1}$ times the weight of all edges in $C$.

$$\sum_{S \in \mathcal{S}} s(S) \le 2\binom{\rho}{r-1} s(C)$$

Let $OPT$ be the minimum possible cut. Now since there are $\binom{\rho}{r}$ cuts, we know the lefthand side here is bounded above by $\binom{\rho}{r} s(OPT)$.

$$\binom{\rho}{r} s(OPT) \le 2\binom{\rho}{r-1} s(C)$$

We can now simplify.

$$s(OPT) \le \frac{2r}{\rho} s(C)$$

But note we are given $\rho = O(t)$. So if we have a $\gamma_t$ approximation for the minimum bisection problem, this means we can find a way to remove $r$ vertices such that the removed weight is at most $O(r/t)\gamma_t$. We can do this again to get a bound on the removal of the blue vertices. This yields a total weight removal of $O(x/t)\gamma_t$. $\qquad \square$

Finally, we can prove Lemma 31, which satisfies the conditions of Lemma 30.

*Proof.* Start by running Lemma 35 on $\mathcal{C}$ to yield $\mathcal{C}_0$. Then we can apply Lemma 37 to yield $\mathcal{C}_1$ with red-blue clustering graph $H_M$ and underlying perfect red-blue matching $M'$. We create $\mathcal{C}'$ by merging components in $H_M$ into clusters. Since the max component size is $\ell$ and the max cluster size in $\mathcal{C}_1$ is $6t$, then the max cluster size in $\mathcal{C}'$ is $6t\ell$. This satisfies condition 2 of being $\mathcal{C}$-good. In addition, it is fair by Proposition 34.

Finally, we utilize the fact that we only moved at most $\ell + 108t^2\ell^2$ vertices from any cluster, and note that we only move vertices of a certain color if we have $O(n)$ of that color in that cluster. Then by Lemma 38, we know we lost at most $O(\ell\gamma_t/t + t\gamma_t/\ell^2)$ fraction of the weight from any cluster. This satisfies the second condition and therefore $\mathcal{C}'$ is $\mathcal{C}$-good. $\qquad \square$

Table 5: Impact of different fairlet decomposition on ratio over original average-linkage in percentage (mean $\pm$ std. dev).

| Samples | 100 | 200 | 400 | 800 | 1600 |
|---|---|---|---|---|---|
| CENSUSGENDER, initial | $74.12 \pm 2.52$ | $76.16 \pm 3.42$ | $74.15 \pm 1.44$ | $70.17 \pm 1.01$ | $65.02 \pm 0.79$ |
| final | $92.32 \pm 2.70$ | $95.75 \pm 0.74$ | $95.68 \pm 0.96$ | $96.61 \pm 0.60$ | $97.45 \pm 0.19$ |
| CENSUSRACE, initial | $65.67 \pm 7.53$ | $65.31 \pm 3.74$ | $61.97 \pm 2.50$ | $59.59 \pm 1.89$ | $56.91 \pm 0.82$ |
| final | $85.38 \pm 1.68$ | $92.98 \pm 1.89$ | $94.99 \pm 0.52$ | $96.86 \pm 0.85$ | $97.24 \pm 0.63$ |
| BANKMARRIAGE, initial | $75.19 \pm 2.53$ | $73.58 \pm 1.05$ | $74.03 \pm 1.33$ | $73.68 \pm 0.59$ | $72.94 \pm 0.63$ |
| final | $93.88 \pm 2.16$ | $96.91 \pm 0.99$ | $96.82 \pm 0.36$ | $97.05 \pm 0.71$ | $97.81 \pm 0.49$ |
| BANKAGE, initial | $77.48 \pm 1.45$ | $78.28 \pm 1.75$ | $76.40 \pm 1.65$ | $75.95 \pm 0.77$ | $75.33 \pm 0.28$ |
| final | $91.26 \pm 2.66$ | $95.74 \pm 2.17$ | $96.45 \pm 1.56$ | $97.31 \pm 1.94$ | $97.84 \pm 0.92$ |

## F   Additional experimental results for revenue

We have conducted experiments on the four datasets for revenue as well. The Table 5 shows the ratio of fair tree built by using average-linkage on different fairlet decompositions. We run Algorithm 1 on the subsamples with Euclidean distances. Then we convert distances into similarity scores using transformation $s(i,j) = \frac{1}{1+d(i,j)}$. We test the performance of the initial random fairlet decomposition and final fairlet decomposition found by Algorithm 1 for revenue objective using the converted similarity scores.

## G   Additional experimental results for multiple colors

We ran experiments with multiple colors and the results are analogous to those in the paper. We tested both Census and Bank datasets, with age as the protected feature. For both datasets we set 4 ranges of age to get 4 colors and used $\alpha = 1/3$. We ran the fairlet decomposition in [3] and compare the fair hierarchical clustering's performance to that of average-linkage. The age ranges and the number of data points belonging to each color are reported in Table 6. Colors are named $\{1, 2, 3, 4\}$ descending with regard to the number of points of the color. The vanilla average-linkage has been found to be unfair: if we take the layer of clusters in the tree that is only one layer higher than the leaves, there is always one cluster with $\alpha > \frac{1}{3}$ for the definition of $\alpha$-capped fairness, showing the tree to be unfair.

Table 6: Age ranges for all four colors for Census and Bank.

| Dataset | Color 1 | Color 2 | Color 3 | Color 4 |
|---|---|---|---|---|
| CENSUSMULTICOLOR | $(26, 38] : 9796$ | $(38, 48] : 7131$ | $(48, +\infty) : 6822$ | $(0, 26] : 6413$ |
| BANKMULTICOLOR | $(30, 38] : 14845$ | $(38, 48] : 12148$ | $(48, +\infty) : 11188$ | $(0, 30] : 7030$ |

As in the main body, in Table 7, we show for each dataset the $\mathrm{ratio}_{\mathrm{value}}$ both at the time of initialization (Initial) and after using the local search algorithm (Final), where $\mathrm{ratio}_{\mathrm{value}}$ is the ratio between the performance of the tree built on top of the fairlets and that of the tree directly built by average-linkage.

Table 7: Impact of Algorithm 1 on $\mathrm{ratio}_{\mathrm{value}}$ in percentage (mean $\pm$ std. dev).

| Samples | 200 | 400 | 800 | 1600 | 3200 | 6400 |
|---|---|---|---|---|---|---|
| CENSUSMULTICOLOR, initial | $88.55 \pm 0.87$ | $88.74 \pm 0.46$ | $88.45 \pm 0.53$ | $88.68 \pm 0.22$ | $88.56 \pm 0.20$ | $88.46 \pm 0.30$ |
| final | $99.01 \pm 0.09$ | $99.41 \pm 0.57$ | $99.87 \pm 0.28$ | $99.80 \pm 0.27$ | $100.00 \pm 0.14$ | $99.88 \pm 0.30$ |
| BANKMULTICOLOR, initial | $90.98 \pm 1.17$ | $91.22 \pm 0.84$ | $91.87 \pm 0.32$ | $91.70 \pm 0.30$ | $91.70 \pm 0.18$ | $91.69 \pm 0.14$ |
| final | $98.78 \pm 0.22$ | $99.34 \pm 0.32$ | $99.48 \pm 0.16$ | $99.71 \pm 0.16$ | $99.80 \pm 0.08$ | $99.84 \pm 0.05$ |

Table 8 shows the performance of trees built by average-linkage based on different fairlets, for Revenue objective. As in the main body, the similarity score between any two points $i, j$ is $s(i,j) = \frac{1}{1+d(i,j)}$. The entries in the table are mean and standard deviation of ratios between the fair tree performance and the vanilla average-linkage tree performance. This ratio was calculated both at time of initialization (Initial) when the fairlets were randomly found, and after Algorithm 1 terminated (Final).

Table 9 shows the run time of Algorithm 1 with multiple colors.

## H   Pseudocode for the cost objective

Table 8: Impact of Algorithm 1 on revenue, in percentage (mean $\pm$ std. dev).

| Samples | 200 | 400 | 800 | 1600 | 3200 |
|---|---|---|---|---|---|
| CENSUSMULTICOLOR, initial | $75.76 \pm 2.86$ | $73.60 \pm 1.77$ | $69.77 \pm 0.56$ | $66.02 \pm 0.95$ | $61.94 \pm 0.61$ |
| final | $92.68 \pm 0.97$ | $94.66 \pm 1.66$ | $96.40 \pm 0.61$ | $97.09 \pm 0.60$ | $97.43 \pm 0.77$ |
| BANKMULTICOLOR, initial | $72.08 \pm 0.98$ | $70.96 \pm 0.69$ | $70.79 \pm 0.72$ | $70.77 \pm 0.49$ | $69.88 \pm 0.53$ |
| final | $94.99 \pm 0.79$ | $95.87 \pm 2.07$ | $97.19 \pm 0.81$ | $97.93 \pm 0.59$ | $98.43 \pm 0.14$ |

Table 9: Average running time of Algorithm 1 in seconds.

| Samples | 200 | 400 | 800 | 1600 | 3200 | 6400 |
|---|---|---|---|---|---|---|
| CENSUSMULTICOLOR | 0.43 | 1.76 | 7.34 | 35.22 | 152.71 | 803.59 |
| BANKMULTICOLOR | 0.43 | 1.45 | 6.77 | 29.64 | 127.29 | 586.08 |

---

**Algorithm 2** Fair hierarchical clustering for cost objective.

---

**Input:** Graph $G$, edge weight $w : E \rightarrow \mathbb{R}$, color $c : V \rightarrow \{\text{red, blue}\}$, parameters $t$ and $\ell$

{Step (A)}
$T \leftarrow$ UNFAIRHC$(G, w)$ {Blackbox unfair clustering that minimizes cost}

{Step (B)}
Let $\mathcal{C} \leftarrow \emptyset$
Do a BFS of $T$, placing visited cluster $C$ in $\mathcal{C}$ if $|C| \leq t$, and not proceeding to $C$'s children

{Step (C)}
$\mathcal{C}_0, C' \leftarrow \emptyset$
**for** $C$ **in** $\mathcal{C}$ **do**
    $C' \leftarrow C' \cup C$
    **if** $|C'| \geq t$ **then**
        Add $C'$ to $\mathcal{C}_0$
        Let $C' \leftarrow \emptyset$
    **end if**
**end for**
If $|C'| > 0$, merge $C'$ into some cluster in $\mathcal{C}_0$

{Step (D)}
**for** $C$ **in** $\mathcal{C}_0$ **do**
    Let $exc(C) \leftarrow$ majority color in $C$
    Let $ex(C) \leftarrow$ difference between majority and minority colors in $C$
**end for**

{Step (E)}
$H_M \leftarrow$ BuildClusteringGraph$(\mathcal{C}_0, ex, exc)$

{Step (F)}
$fV \leftarrow$ FixUnmatchedVertices$(\mathcal{C}_0, H_M, ex, exc)$

{Step (G)}
$\mathcal{C}' \leftarrow$ ConstructClustering$(\mathcal{C}_0, ex, exc, fV)$
**return** $\mathcal{C}'$

---

**Algorithm 3** BuildClusteringGraph $(\mathcal{C}_0, ex, exc)$

---

$H_M \leftarrow (V_M = \mathcal{C}_0, E_M = \emptyset)$
Let $C_i \in V_M$ be any vertex
Let $\ell \leftarrow n^{1/3}\sqrt{\log n}$
**while** $\exists$ an unvisited $C_j \in V_M$ such that $exc(C_j) \neq exc(C_i)$ **do**
    Add $(C_i, C_j)$ to $E_M$
    Swap labels $C_i$ and $C_j$ if $ex(C_j) > ex(C_i)$
    Let $ex(C_i) \leftarrow ex(C_i) - ex(C_j)$
    **if** $ex(C_i) < \ell$ or $|component(C_i)| \geq \ell$ **then**
        Reassign starting point $C_i$ to an unvisited vertex in $V_M$
    **end if**
**end while**
**return** $H_M$

---

**Algorithm 4** FixUnmatchedVertices$(\mathcal{C}_0, H_M, ex, exc)$

---

Let $\ell \leftarrow n^{1/3}\sqrt{\log n}$
**for** $C \in \mathcal{C}_0 \setminus V_M$ **do**
    Let $fV(C, \text{red}), fV(C, \text{blue}) \leftarrow m^2/\ell^2$
**end for**
**for** $i$ from 1 **to** $108t^2/\ell^3$ **do**
    **for** each $k$ component in $H_M$ **do**
        **for** $p$ **in** a BFS of $k$ **do**
            Let $ch \leftarrow$ some child of $p$
            $fV(p, exc(p)) \leftarrow fV(p, exc(p)) + \ell$
            $ex(p) \leftarrow ex(p) - \ell$
            $fV(ch, exc(ch)) \leftarrow fV(ch, exc(ch)) + \ell$
            $ex(ch) \leftarrow ex(ch) - \ell$
            **if** # matches between $p$ and $ch < \ell$ **then**
                Remove $(p, ch)$ from $E_M$ {This creates a new component}
            **end if**
        **end for**
    **end for**
**end for**
**return** $fV$

---

**Algorithm 5** ConstructClustering$(\mathcal{C}_0, ex, exc, fV)$

---

Let $\mathcal{C}', R \leftarrow \emptyset$
**for** $C$ in $\mathcal{C}_0$ **do**
    **for** $c$ **in** {red, blue} **do**
        Let $f = fV(C, c)$
        Let $C_f = \{v \in C : c(v) = c\}$
        Create the transformed graph $L$ from $C_f$ {Described in the proof of Lemma 38}
        $C' \leftarrow$ MINWEIGHTBISECTION$(L)$ {Blackbox, returns isolated $C_f$ vertices}
        $C \leftarrow C \setminus C'$
        $R \leftarrow R \cup C'$
        $ex(C) \leftarrow ex(C) - |C'|$
    **end for**
**end for**
**for** $C \in \mathcal{C}_0$ **do**
    Let $S \subset R$ such that $|S| = ex(C)$ with no vertices of color $exc(C)$
    $C = C \cup S$
    $R \leftarrow R \setminus S$
    Add $C$ to $\mathcal{C}'$
**end for**
**return** $\mathcal{C}'$

---