[Reviews · NeurIPS 2020]

Review 1

Summary and Contributions: This paper extends the fairlets machinery to hierarchical clustering under different objectives (revenue, value and cost).Theoretical results have been derived and polynomial time algorithms have been proposed for the objectives of revenue and value. Empirical evaluation has been conducted to verify the effectiveness and efficiency of the proposed approach.

Strengths: Fair clustering is an important topic in the field of machine learning. The paper has analyzed quality guarantees and proposed polynomial time algorithms for optimizing revenue and value with fairness.

Weaknesses: Some important definitions, such as beta approximation, should have been added in the paper to make it self-contained. As mentioned by the authors, algorithm proposed by ref [14] is also a fair hierarchical algorithm.Its results may be compared to assess the effectiveness and efficiency of the proposed approach. ------- After Authors feedback------- I would like to thank authors for addressing the comments. I have read the author response and reviewers’ discussions.The technical novelty of the paper has been made clearer during the discussions between the reviewers. So I increase my score to 7.

Correctness: Correct.

Clarity: Well written.

Relation to Prior Work: Clear.

Reproducibility: Yes

Additional Feedback: Some important definitions, such as beta approximation, should have been included in the paper to make it self-contained. Algorithm proposed by ref [14] may be compared to assess the effectiveness and efficiency of the proposed approach.


Review 2

Summary and Contributions: This paper applies established fairness notions such as disparate impact and representational fairness to hierarchical clustering objectives. The authors show how to extend the fairlet technique (essentially finding micro-clusters that (i) are a lower bound on the general cost, (ii) satisfy the desired fairness notion (iii) are easier to compute) to various objectives proposed for graph clustering. The authors show that this allows them to preserve state of the art approximation guarantees for the various objectives that they consider.

Strengths: Straight off the bat, this is one of the more convincing fairness papers I have seen submitted to this year's NeurIPS. I like that this paper sticks to established notions of fairness and how the authors show how fairlets can be used here. If there exists an algorithmic technique in the field of fair algorithms, then fairlets have a strong claim to the title, as they have received by far the most numerous attention (in my opinion). The results themselves are solid. I would like to highlight the final result that the authors gave, being the extension of the DasGupta objective. I think this result is highly interesting, and while I understand that the authors buried it somewhat for the sake of space, I think that this is arguably the most intersting contribution. Technically, this paper is also non-trivial. Indeed, if this paper bridges a gap between people usually not interested in algorithms and the more theoretically minded part of the NeurIPS community, then this is a welcome side effect.

Weaknesses: I wish the experiments had been put into the supplementary material and that the paper had given more room for the actually interesting results. Then again, I guess this is expected of a NeurIPS paper, so who am I to argue.

Correctness: As far as I can tell, yes. I only thoroughly checked theorem 17, which I believe is the most interesting result. The other results are less surprising, and therefore plausible.

Clarity: Yes.

Relation to Prior Work: Mostly, related work is solid. It should be noted that Kleindessner et al. do not actually give a linear time approximation for the same problem (in fact, they only give a result for what is essentially k-center under a matroid constraint). The authors also could have cited Schmidt et al. Fair Coresets and Streaming Algorithms for Fair k-means (WAOA 2019).

Reproducibility: Yes

Additional Feedback: I think this is the strongest paper in my pile, and definitely one of the stronger fairness papers. I would like to see it accepted.


Review 3

Summary and Contributions: The paper studies the hierarchical clustering in which the goal is to recursively partition the input to minimize certain objective functions with group fairness requirement. In group fairness requirement, each cluster has at most alpha fraction of its point from a same group.

Strengths: The paper provides simple algorithms with provable guarantees for fair hierarchical clustering with respect to three standard objective functions studied for this problem. In general, it has a solid contribution and expands the understanding of fair clustering.

Weaknesses: The techniques seem standard (in the area of fair clustering) and the paper has limited novelty in terms of of technical contributions. While hierarchical clustering is a natural problem, and fair clustering is a well-motivated problem, it will be helpful if authors provide more motivations/justifications why this notion of fairness is useful for hierarchical clustering.

Correctness: The general proof sketch and results seem correct (I have not verified all details in the appendix) and experimental results are also reasonable and helpful.

Clarity: The paper is well written and has a good flow.

Relation to Prior Work: The paper has discussed related works comprehensively and has done a fair comparison. I have some suggestion in the additional feedback part.

Reproducibility: Yes

Additional Feedback: Line 68: Kleindessner et al. designed an algorithm for k-center with different type of fairness requirement. Instead of balancing different colors in each cluster, the goal is to pick centers (proportionally) from different colors. It is basically k-center under partition matroid. Line 69-70: In a(n almost) concurrent work, the fair correlation was also studied by Ahamdi et al. Line 131: Bounded representation: with binary colors, it is the same as balance. Bera et al. and Bercea et al. have also considered a generalization of this notion prior to Ahmadian et al. How does bounded representation compare to them? On the cost of essentially fair clusterings by Bercea et al. (2018) Fair Algorithms for Clustering by Bera et al. (2019) Line 154 and Line 188 (Theorem 8): what is an upper bound on the value of m_f in terms of alpha (for general values of alpha)? Algorithm 1 – step 2: how to find an initial solution? Line 270: For for Line 291: obseve -> observe Line 300: usint -> using ------- After Authors feedback I decided to increase my score to "accept" but as Reviewer 2 I also suggest authors to provide a better presentation of their theoretical contributions (maybe in the introduction).

[Author Response · NeurIPS 2020]

We thank all the reviewers for their time, feedback, and insightful comments.

**Review #1.**

$\beta$-approximation: By $\beta$-approximation, we mean an algorithm that approximates the optimum within a multiplicative factor of $\beta$. We will add a formal definition in the revision.

Comparison to [14]: This manuscript appeared on arXiv contemporaneously with our submission to NeurIPS.

Compared to our results, the main takeaways from [14] are: (i) their algorithms are heuristic and therefore do not provably optimize for any objective such as value or revenue, and (ii) their algorithms are not provably fair under our fairness constraints. Thus, unlike ours, their work does not seem to offer any formal guarantees.

For an experimental comparison, we tried the following two routes during the rebuttal process:

(i) We reached out to the authors for a pointer to their code. The authors responded that their code is currently unavailable as it is being revised based on the feedback they received.

(ii) We implemented their algorithms from scratch. During this process, we realized that replicating their results is difficult due to the lack of a completely specified procedure for choosing the hyperparameters of their algorithm $(\alpha_0, \beta_0, \theta_1, \theta_2)$. For these reasons, to ensure a fair comparison with their work, we postpone the experimental evaluation of their algorithm pending the release of the new implementation by authors.

**Review #2.**

Thank you for the positive comments. We will address your suggestions and include the additional citation in the revision.

**Review #3.**

Apologies for the typos and thank you for the careful reading.

Kleindessner et al. designed an algorithm for k-center with different type of fairness requirement. Instead of balancing different colors in each cluster, the goal is to pick centers (proportionally) from different colors. It is basically k-center under partition matroid.

Thank you for articulating this clearly; we will add this in the revision.

Both the bounded representation and the representations from Bera et al. and Bercea et al. generalize the problem of "balance". How do these different representations compare?

Bera et al. and Bercea et al. both introduce a generalized version of the constraint from [Ahmadian et al. 2019]. For each color, they specify the upper and lower bounds for their fractional representation in each cluster. Our results continue to hold even for this general constraints, as long as we can find a fairlet decomposition satisfying the conditions of Theorems 8 and 10. We will add this remark to the revision.

(Theorem 8): what is an upper bound on the value of $m_f$ in terms of $\alpha$ (for general values of $\alpha$)?

For two colors with general $\alpha = r/(b + r)$, where $r \geq b$, we have $m_f \leq b + r$ [Chierichetti et al. 2019]. For multiple colors with $\alpha = 1/t$, we have $m_f \leq 2t - 1$. Note that the bound for multiple colors for general alpha is an open question.

Algorithm 1 – step 2: how to find an initial solution?

For two colors with general $\alpha = r/(b + r)$, we use the fairlet decomposition method proposed in [Chierichetti et al. 2019]. For multiple colors with $\alpha = 1/t$, we use the method proposed in Lemma 24 in the Supplementary Material.

In the revision, will also add further justifications for studying fairness in a hierarchical clustering setting.

[Meta-Review · NeurIPS 2020]

The paper considers hierarchical clustering with fairness constraints and show the concept of fairlet decomposition extends to fair hierarchical clustering as well. The notion of fairness is important though it would be good if the authors improve the motivation section of this work. Overall, a solid theoretical work.